# LEARNING STRUCTURED DEPENDENCIES USING GENERATIVE COMPUTATIONAL UNITS

## ABSTRACT

The ability of neural networks to generalize from data is fundamentally shaped by the design of their computational units. Common examples include the perceptron and radial basis function (RBF) units, each of which provides useful inductive biases. In this work, we introduce a new computational unit, the generative matching unit (GMU), which is designed to naturally capture structured dependencies in data. Each GMU contains an internal generative model that infers latent parameters specific to an input instance $X$, and then outputs a non-linear function of the generative error using these parameters. By incorporating generative mechanisms into the unit itself, GMUs offer a complementary approach to existing computational units. In this work, we focus on linear GMUs, where the internal generative models are linear latent variable models, yielding a function form that shares some similarities with RBF units. We show that linear GMUs are universal approximators like RBFs, while being able to convey richer information and lessen the impact of the curse of dimensionality compared to RBFs. Like perceptrons and RBF units, a linear GMU has its set of weights and biases, and has a closed-form analytical expression, enabling fast computation. To evaluate the performance of linear GMUs, we conduct a set of comprehensive experiments and compare them to multi-layer perceptrons (MLPs), RBF networks, and ResNets. We construct GMU-ResNets, where the first feedforward layer is replaced by GMUs, and test on 27 tabular datasets, observing improved generalization over standard ResNets and competitive performance with other benchmarks. We also construct GMU-CNNs, which contain convolutional GMUs in their first layer. Across five vision datasets, GMU-CNNs exhibit better generalization and significantly better robustness to test-time corruptions. We also empirically compare linear GMUs, to benchmark networks across more than 30 synthetic classification tasks, encompassing both structured and unstructured data distributions. We find that GMUs consistently demonstrate superior generalization to out-of-distribution samples, especially for the structured cases.

## 1 INTRODUCTION

Computational units serve as the fundamental building blocks of deep learning architectures, shaping the network's ability to process and generalize. While the universal approximation theorem provides insights into the representational capacity of multilayered architectures, the behavior of individual computational units plays a crucial role in determining the network's extrapolative and interpolative properties. Understanding these intrinsic characteristics at a granular level remains essential for analyzing how a model generalizes beyond its training data and adapts to new inputs.

A widely used computational block in neural architectures is the perceptron unit, which introduces non-linearity essential for learning complex functions. This unit has remained foundational across various modern architectures, including transformer variants (Lin et al., 2022) and convolutional networks (Li et al., 2021). Multi-layered perceptron (MLP) networks are known to be universal approximators, capable of learning intricate mappings given sufficient complexity and data (Hornik et al., 1989). Another example of a classical computational unit is the Radial Basis Function (RBF) unit, which also has universal approximation capabilities. Recent years have seen other computational units being introduced, such as self-attention, Kolmogorov-Arnold Networks (Liu et al., 2024) and Capsule networks (Patrick et al., 2022; Hinton et al., 2018). Alternatively, there have been approaches that preserve the network computational structure, but dynamically adapt the parameters. Some examples of that include Hypernetworks, where alternative network are used to adjust weights (Ha

et al., 2016; Chen et al., 2020), deformable convolution variants (Dai et al., 2017), where both the convolution weights and location perturbations are learnt simultaneously and spatial transformer networks, where a differentiable transformation is learnt on the image before forward it to a standard neural network.

In our work, we undertake a different approach. We use the Bayesian framework to motivate the design of new computational units, rethinking their properties through a probabilistic lens. Specifically, given the input $X$ and target label $Y$, we seek to characterize the class of joint distributions $P(X, Y)$ under which a single layered network with individual units, coupled with common loss functions like cross-entropy, can achieve *Bayes-optimal* classification. This is equivalent to identifying scenarios when a single-layer network with certain computation units can exactly recover the underlying class conditionals $P(Y|X)$.

Through this framework, we observe that perceptron units attain Bayes-optimal classification when the underlying data distribution naturally factorizes into conditionally independent exponential distributions. However, real-world data often exhibits complex inter-dimensional dependencies beyond this simplified structure. To emulate this scenario, we use Structural Causal Models (SCMs), which offer a principled framework for modeling causal mechanisms in data generation, extending probabilistic models to incorporate causal relationships (Neuberg, 2003). To that end, we introduce a new generative framework called *Class-wise Latent Structural Causal Model (CL-SCM)*, where each class has a distinct structural causal model governing input generation. When the data generation model is a CL-SCM, we identify a novel computational unit that can exactly recover the underlying class conditionals.

We call our proposed computational unit *Generative Matching Units (GMUs)*, which function fundamentally differently from traditional computational units. Instead of directly transforming input features, GMUs reconstruct inputs using an internal generative model, minimizing reconstruction error within their framework. This bridges the gap between local feedforward units and holistic structured generative modeling. Importantly, linear GMUs, which is explored in this work, has a closed form analytical expression, enabling fast and efficient inference. GMUs thus offer a promising approach for capturing complex data structures in relevant problems. We outline the contributions of our work as follows:

1. We show the correspondence between computational units and Bayes Optimality, and show that under structured generative settings, a new computational unit emerges. Based on these findings we introduce the Generative Matching Unit (GMU) and linear GMU variants.
2. We show that GMU networks can be universal approximators under mild constraints (which are satisfied by linear GMUs) while simultaneously being able to capture richer information in high-dimensional spaces than RBF networks, lessening the impact of the curse of dimensionality.
3. To study linear GMUs, we conduct an exhaustive comparison of performance between GMU-ResNets and ResNets on 27 tabular datasets from openML. We find that in the majority of cases GMU-ResNets generalize better than ResNets, and in some cases they yield state-of-the-art results, when compared to well-known benchmarks.
4. We test GMU-CNN variants on five standard vision datasets. We find that GMU-CNNs significantly outperform their vanilla CNN counterparts in terms of robustness to test-time corruptions.
5. GMUs show significantly better out-of-distribution generalization in diverse synthetic data settings.

## 2 BAYES OPTIMALITY OF COMPUTATIONAL UNITS

In this section, we study when computational units can recover the true conditional probabilities, thereby achieving Bayes optimal classification. We begin with a simple conditionally independent setting for perceptrons, and then extend to more structured cases, including class-wise latent models that can be viewed as instances of switching causal models (Willig et al., 2025), which motivates our new generative computational unit. Given random variables $X$ and $Y$ denoting inputs and labels, we consider samples $S = \{(X_1, Y_1), \ldots, (X_n, Y_n)\}$ from $P(X, Y)$. We highlight the connection between the ground-truth conditional $P(Y \mid X)$ and the output of a single-layer network with specific computational units under different sampling processes, as outlined in Figure 1, beginning with the exponential conditionally independent sampling process.

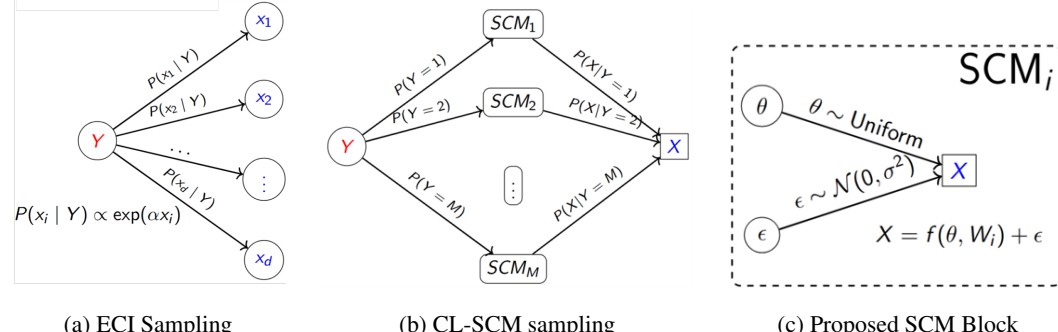

| (a) ECI Sampling | (b) CL-SCM sampling | (c) Proposed SCM Block |

**Figure 1:** Comparing ground truth data generation models (a) Exponential Conditionally Independent (ECI) Sampling, which samples every input dimension $x_i$ independently given the label $Y$ and (b) Class-wise latent structural causal model based sampling (CL-SCM), which maintains a separate SCM for each class. (c) outlines the SCM in the CL-SCM approach, which can emulate more complex structural dependencies in the data.

**Definition 1** (Exponential Conditionally Independent (ECI) Sampling). *Let $X = [x_1, x_2, \ldots, x_d] \in \mathbb{R}^d$ and class label $Y \in \{1, \ldots, M\}$. The data is sampled such that $P(Y|X) \propto P(Y) \prod_{j=1}^{d} P(x_j|Y)$, where each feature $x_j$ is conditionally independent given $Y$, and follows the exponential form $P(x_j|Y = i) = \frac{1}{Z_{ij}} \exp(\alpha_{ij} x_j)$, where $Z_{ij}$ is a normalization factor.*

A single-layer perceptron trained with cross-entropy recovers $P(Y \mid X)$ under the ECI assumption.
**Proposition 1.** *Let the random variables $(X, Y)$ follow the ECI sampling process, and let $S = \{(X_1, Y_1), \ldots, (X_n, Y_n)\}$ be i.i.d. samples from this distribution. Consider a single-layer neural network with computational units $\mathrm{pu}_i(X, \mathbf{W}, \mathbf{b}) := \sum_{j=1}^{d} W_{ij} x_j + b_i, \quad i \in \{1, \ldots, M\}$, followed by a softmax layer producing output $P_{\mathrm{soft}}(Y|X, \mathbf{W}, \mathbf{b})$. Let $\mathcal{L}_{\mathrm{CE}}$ denote the empirical cross-entropy loss and assume it is uniformly bounded over the support of $P(Y|X)$. Then, for every $(\mathbf{W}^*, \mathbf{b}^*) \in \arg\min_{\mathbf{W}, \mathbf{b}} \mathcal{L}_{\mathrm{CE}}(S), \lim_{n \to \infty} \mathrm{KL}\left(P(Y|X) \,\|\, P_{\mathrm{soft}}(Y|X, \mathbf{W}^*, \mathbf{b}^*)\right) = 0$ almost surely.*

While the ECI sampling assumes conditional independence between input dimensions, real-world data often exhibits intricate dependencies that standard models fail to capture. To address this, we introduce a structured sampling process that encodes these inter-dimensional relationships, called Class-wise Latent Structural Causal Model (SCM) sampling.
**Definition 2** (Class-wise Latent SCM (CL-SCM) Sampling). *Let $X = [x_1, x_2, \ldots, x_d] \in \mathbb{R}^d$ be generated by one of $M$ latent structural causal models (SCMs) corresponding to each class $Y \in \{1, \ldots, M\}$: $X = f(\theta, W_Y) + \epsilon, \quad \epsilon \sim \mathcal{N}(0, \sigma^2 I), \theta \sim \mathcal{U}([-1, 1]^k). \quad i \in \{1, \ldots, M\}$, where $\theta \in \mathbb{R}^k$ represents latent confounders influencing $X$, and $W_i$ corresponds to the structural parameters of the causal model. $\mathcal{U}$ denotes the uniform distribution.*

The class-wise latent SCM (CL-SCM) can be viewed as a special case of switching SCMs (Willig et al., 2025), where different causal mechanisms are activated depending on context and the active mechanism is determined by the class label $Y$. In the CL-SCM case, we show a new computational unit emerges that recovers the true posterior, but only under a MAP approximation.
**Proposition 2.** *Let the random variables $(X, Y)$ follow the CL-SCM sampling process with equal class priors, and let $S = \{(X_1, Y_1), \ldots, (X_n, Y_n)\}$ be i.i.d. samples from this distribution. Consider a single-layer neural network with computational units $\mathrm{g}_i(X, \mathbf{W}, \sigma) := -\frac{1}{\sigma^2} \min_\theta \mathbb{E}[\|f(\theta, W_i) - X\|^2]$ for $i \in \{1, \ldots, M\}$, followed by a softmax layer producing output $P_{\mathrm{soft}}(Y \mid X, \mathbf{W}, \sigma)$. Let $\mathcal{L}_{\mathrm{CE}}$ denote the empirical cross-entropy loss and assume it is uniformly bounded over the support of $P(Y \mid X)$. Then, for every $(\mathbf{W}^*, \sigma^*) \in \arg\min_{\mathbf{W}, \sigma} \mathcal{L}_{\mathrm{CE}}(S)$, we have $\lim_{n \to \infty} \mathrm{KL}(P(Y \mid X) \,\|\, P_{\mathrm{soft}}(Y \mid X, \mathbf{W}^*, \sigma^*)) = 0$ almost surely.*

The computational unit $g_i$ explicitly performs an internal optimization for every input instance $X$, using an internal generative model $f(\theta, W_i)$ to reconstruct $X$. This directly motivates the need for generative computational units, which we outline in the following sections.
**Remark 1.** *However, we observe the unit $g_i$ may not be directly adaptable to real architectures. When active, its output remains close to zero, whereas when inactive, it can become significantly negative. To appropriately adjust its range, the final form of the GMU incorporates an additional*

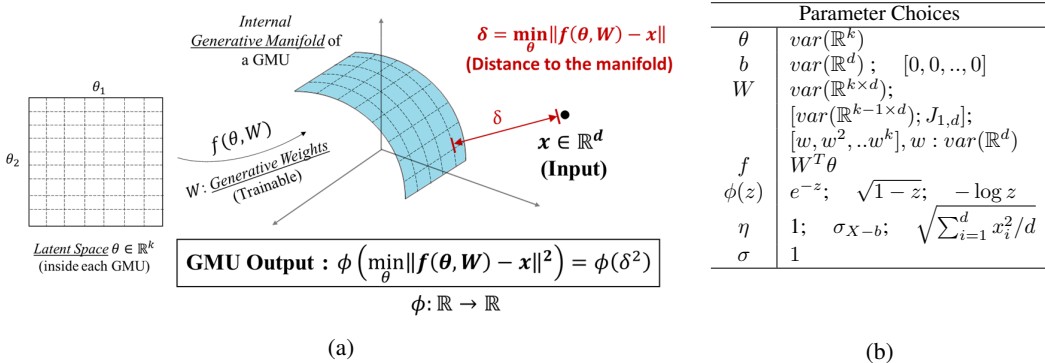

**Figure 2:** (a) A conceptual depiction of a single generative matching unit and (b) Table showing potential parameter choices for the GMU function in equation 1.

*activation function $\phi$, ensuring that the unit maintains a positive activation when it effectively predicts the input and approaches zero when it does not.*

## 3   GENERATIVE MATCHING UNITS: VARIANTS AND THEORETICAL RESULTS

We first define the most general form of a GMU as follows.

**Definition 3.** *(**Generative Matching Unit:**) Let the input to the unit be $X = [x_1, x_2, .., x_d] \in \mathbb{R}^d$, where $x_i \in \mathbb{R}$. Consider a function family $\mathcal{F}$, such that every function $f : \mathbb{R}^k \to \mathbb{R}^d \in \mathcal{F}$ can be parameterized as $f(\theta, W) + b$, where $\theta \in \mathbb{R}^k$, $W \in \mathbb{R}^{k \times d}$, and $b \in \mathbb{R}^{1 \times d}$. $W$ and $b$ represent the generative weights and biases of the unit, and $\theta$ represents the latent generating variables. Then, a GMU of order $k$ computes:*

$$gmu(X) = \phi\left(\frac{1}{\sigma^2 d} \min_{\theta \in \mathbb{R}^k} \left\| \frac{X-b}{\eta} - f(\theta, W) \right\|^2\right) \tag{1}$$

$\phi : \mathbb{R} \to \mathbb{R}$ *is the GMU's activation function. $\eta$ represents an optional normalization measure to ensure that $\frac{1}{d} \min_{\theta \in \mathbb{R}^k} \|\frac{X-b}{\eta} - (f(\theta, W))\|^2$ is bounded. Lastly, $\sigma$ represents an optional smoothing factor.*

We outline all parameter and function choices in equation 1 tested in this paper in Figure 2(b) . In Figure 2(b), all real number based entries of the form $var(\mathbb{R}^{p \times q})$ denote tensors of size $p \times q$ where all entries are real variables subject to gradient descent. Lastly, $J_{p,q}$ denotes a constant matrix of dimensions $p \times q$, where every element is equal to 1. We next discuss the linear GMU variant that is tested in this work. For notational ease, we denote a GMU of order $k$ by GMU($k$).

**Remark 2.** *We provide a conceptual depiction of the computation inside a GMU in Figure 2(a). Each GMU's internal generative model can be parameterized by a manifold over the space of the input $X$. Subsequently, for every input instance $x \in \mathbb{R}^d$, a GMU effectively estimates the distance between its internal manifold and $x$, and returns a function ($\phi$) of this distance. Ideally, $\phi$ should be chosen such that lower distances yield higher activations and vice-versa.*

**Remark 3.** *($\mu$-RBF units) Radial Basis Functions (Buhmann, 2000; Powell, 1987) are a special case of equation 1 . Specifically, when $k = 0$, $\eta = 1$, $\phi(z) = e^{-z^2}$, and $f(\theta, W) = 0$, we have $gmu(X) = \phi\left(\frac{1}{\sigma^2 d} \|X - b\|^2\right)$. GMUs contain an additional averaging across dimensions yielding the $d$ term in the denominator which RBFs don't have. Furthermore, in our experiments we find that performance is more stable when we set $\sigma = 1$. We denote them as $\mu$-RBF networks, and these units specifically as $\mu$-RBF units.*

### 3.1   LINEAR GMUS

In this work, we restrict $f$ to linear functions $W^T \theta$ over the latent generating variables. The resulting generative structure represents a specific case of linear latent variable models (LLVMs), which have been studied in literature in the context of generative modeling (Reilly, 2025). Here we set $\sigma = 1$, yielding the following expression for a GMU of order $k$:

$$gmu(X) = \phi\left(\frac{1}{d} \min_{\theta \in \mathbb{R}^k} \left\| \frac{X-b}{\eta} - W^T \theta \right\|^2\right) \tag{2}$$

Linear $f$ allows for closed form expressions of equation 2 as it becomes a least squares regression problem. This eventually yields the following final expression:

$$\text{gmu}(X) = \phi\left(\frac{1}{d}\left\|\frac{X-b}{\eta} - W^T(W^T)^\dagger\left(\frac{X-b}{\eta}\right)\right\|^2\right),$$ (3)

where $A^\dagger$ denotes the pseudo-inverse of A. We denote these as linear GMUs. Note that the generative manifold of linear GMUs is therefore a linear subspace of dimensionality $k$. Geometrically, when $\eta = 1$, $\min_{\theta \in \mathbb{R}^k}\left\|\frac{X-b}{\eta} - W^T\theta\right\|^2$ becomes the distance between the input point $X$ and a linear subspace parameterized by $W^T\theta + b$ (manifold in Figure 2(a)). Lastly, in our experiments, we set $\sigma = 1$, as we find that the $1/d$ term is enough to avoid collapsing gradients, and learning is stable. We also find that $\sigma = 1$ yields more performant GMUs.

## 3.2 ARE GMUS UNIVERSAL APPROXIMATORS?

We study the universal approximation abilities of two layered GMU-MLPs, which consists of a layer containing multiple GMUs followed by a linear perceptron layer. First, let us define the set of functions $L^p(\mathbb{R}^d)$ such that every $f \in L^p(\mathbb{R}^d)$, where $f : \mathbb{R}^d \to \mathbb{R}$, is $p^{th}$ power integrable, bounded, continuous and continuous with compact support. $L^p(\mathbb{R}^d)$ encompasses the set of all such functions which satisfy these constraints. This leads to the universal approximation theorem for GMUs, as an extension of the result in Park and Sandberg (1991).

**Proposition 3.** *(from Park and Sandberg (1991)) We are given a GMU-MLP, with GMU units in equation 1 of arbtrary $k$ specified as: $\eta = 1$ and $\exists W s.t. f(\theta, W) = C \forall \theta$ (C is any constant) and $\phi(z)$ is any integrable bounded function such that $\int \phi(x)dx \neq 0$. Then, this GMU-MLP can approximate any function $f \in L^p(\mathbb{R}^d)$.*

Note that linear GMUs satisfy the constraints of Proposition 3, and thus are universal approximators.

## 3.3 COMPARING LINEAR GMUS WITH RBFS

Linear GMUs merit a direct comparison to RBFs in some ways, because both GMUs and RBFs compute distanced internally, and return a function of this distance. For RBFs, it is the Euclidean distance $||X_1 - X_2||$, whereas for linear GMUs of order $k$, it is the distance between a $k$-dimensional linear subspace and a point. We call this the $k$-subspace distance, and it is formally defined as follows:

**Definition 4** ($k$-Subspace Distance). *The $k$-subspace distance between two points $X_1, X_2 \in \mathbb{R}^d$ with respect to a generative weight matrix $W \in \mathbb{R}^{k \times d}$ is given by:*

$$S_{k,W}(X_1, X_2) = \min_{\theta \in \mathbb{R}^k}\left\|X_2 - (W^T\theta + X_1)\right\|.$$

*This measures the closest distance between $X_2$ and the $k$-dimensional affine subspace generated by $W$ which passes through $X_1$.*

We note that the linear GMU function, as seen in equation 2, computes a $k$-subspace distance. When $k = 0$, the above just computes Euclidean distance, which is used in RBFs. As both methods rely on distance measures, the curse of dimensionality becomes a relevant concern.

**Curse of Dimensionality:** As dimensionality increases, distance-based metrics tend to lose their discriminative power. In high-dimensional spaces, most points become nearly equidistant from one another, causing similarity measures to collapse (Xia et al., 2015). The effectiveness of Euclidean distance diminishes as variability in measured distances decreases, making tasks like nearest-neighbor classification or kernel methods more challenging.

This naturally raises the question: is the $k$-subspace distance inherently more robust to the curse of dimensionality than Euclidean distance? To address this, we introduce the concept of the *coefficient of Variation* (CV), which quantifies how informative a similarity measure remains in high-dimensional spaces.

**Definition 5.** *(Coefficient of Variation) We are given a similarity measure $S(X_1, X_2)$, where $X_1, X_2 \in \mathbb{R}^d$. Let $I_d$ represent the identity matrix of size $d \times d$. Then, the coefficient of variation (CV) $\gamma_S(d)$ of S in d-dimensional space is given by:*

$$\gamma_S(d) = \frac{\sigma_{X_1, X_2 \sim \mathcal{N}(0, I_d)}[S(X_1, X_2)]}{\mu_{X_1, X_2 \sim \mathcal{N}(0, I_d)}[S(X_1, X_2)]},$$ (4)

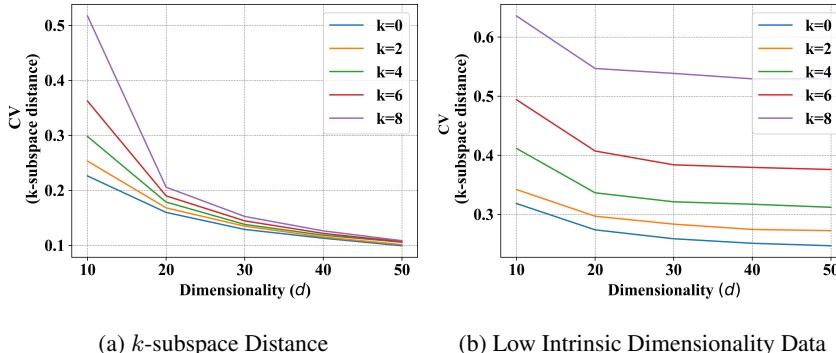

(a) $k$-subspace Distance

(b) Low Intrinsic Dimensionality Data

**Figure 3:** CV v/s Dimensionality of various distance measures. We compare $k$-subspace distance used by GMUs to Euclidean distance ($k = 0$) which is used by units such as RBFs.

*where $\sigma[.]$ and $\mu[.]$ denote the standard deviation and the mean of the random variables within, given the distributions of $X_1$ and $X_2$ below.*

This ratio highlights the degree of variation in a similarity measure. A small $\gamma_S(d)$ indicates low variability and suggests that the measure loses its ability to differentiate between points as dimensionality increases. In contrast, a high CV signifies greater variability, preserving meaningful distinctions between points.

**Theoretical Results:** To investigate whether $k$-subspace distance is more resistant to the curse of dimensionality, we compare the CV for Euclidean distance ($\gamma_E$) and $k$-subspace distance ($\gamma_{S_{k,W}}$) theoretically. We have the following results.

**Theorem 4.** *Let $E(X_1, X_2) = \|X_1 - X_2\|$, we can show that $\gamma_E(d + 1) < \gamma_E(d)$.*

**Corollary 4.1.** *We are given the $k$-subspace distance $S_{k,W}(X_1, X_2)$ from $X_1$ to $X_2$. With this, first, we note that $\gamma_{S_{0,W}}(d) = \gamma_E(d)$, where $E$ denotes the Euclidean distance. Then, we have that $\gamma_{S_{k+1,W}}(d) > \gamma_{S_{k,W}}(d)$, and thus $\gamma_{S_{k,W}}(d) > \gamma_E(d)$.*

**Proposition 4.** *We consider the case where the RV $X$ has low intrinsic dimensionality. Let $X \in \mathbb{R}^d$ be generated as: $X = \sum_{i=1}^{k_{ID}} a_i v_i$, $a_i \sim \mathcal{N}(0, 1)$, where $\{v_i\}_{i=1}^{k_{ID}}$ are orthonormal vectors in $\mathbb{R}^d$, and $k_{ID} < d$ is the intrinsic dimensionality of $X$. Let us set $W = [v_1, .., v_k]$ to be the first $k$ orthonormal vectors. Then, we have that $\gamma_E(d) = \gamma_E(k_{ID})$ and $\gamma_{S_{k,W}}(d) = \gamma_E(k_{ID} - k)$.*

**Remark 5.** *These results provide interesting insights. Proposition 4 reinforces the curse of dimensionality, showing that Euclidean distance loses information as dimensionality increases. In contrast, Proposition 4.1 demonstrates that $k$-subspace distance retains a strictly greater CV, enabling more robust differentiation in high-dimensional spaces. Moreover, higher-order GMUs extract richer structural information as $k$ increases. Lastly, Proposition 4 highlights that when data in high dimensional ambient space has a low intrinsic dimension, which is common in real-world datasets, $k$-subspace distance can maintain a high CV.*

**Empirical verification:** We conduct a series of experiments to estimate the CV of both Euclidean distance and $k$-subspace distance, verifying the theoretical results from Corollary 4.1 and Theorem 4. For the first experiment, we simulate $X_1, X_2 \sim \mathcal{N}(0, I_d)$ and compute the CVs $\gamma_E(d)$ and $\gamma_{S_{k,W}}(d)$ as functions of dimensionality $d$. The results, summarized in Figure 3(a), confirm that $k$-subspace distance consistently exhibits higher CV compared to Euclidean distance ($k = 0$), reinforcing the idea that GMUs can capture greater structural variability in high-dimensional settings.

Next, we conduct an experiment on data distributions with low intrinsic dimensionality (ID), a common property of real datasets. We hypothesize that when data lies in a low-dimensional subspace within $\mathbb{R}^d$, appropriately chosen weights $W$ can enhance the CV of $k$-subspace distance. To test this, we generate inputs constrained to 10-dimensional linear subspaces within ambient spaces where $d$ varies from 10 to 50, setting the first $k$ rows of $W$ to match the subspace's generating vectors. As shown in Figure 3(b), $k$-subspace distance yields substantial CV improvements for larger $k$, when the data distributions are of low ID. Unlike the previous case, these improvements persist even when the ambient data dimensionality increases significantly.

---

**Algorithm 1** Sparse Linear Structure Sampling

---

1: **Setup:** $X \in \mathbb{R}^d$, labels $y \in \{1, \ldots, C\}$, $W = [W_1, W_2, .., W_{N_{\text{total}}}] \in \mathbb{R}^{N_{\text{total}} \times d}$
2: **For each class $y$:** Select up to $N_{\max}$ active latent variables from $\{\theta_i\}_{i=1}^{N_{\text{total}}}$
3: **Generative Process:**
4:      Sample $y \sim \text{Unif}\{1, \ldots, C\}$
5:      Compute $x$:

$$x = \sum_{i=1}^{N_y} W_{g_y(i)}^T \theta_{g_y(i)} + \epsilon$$

6: where $\epsilon \sim \mathcal{N}(0, \sigma^2 I_d)$

---

## 4 EXPERIMENTS

We provide experiments on both synthetic and real datasets. The synthetic experiments are designed to isolate and stress-test specific structural properties of GMUs, while the real-data experiments demonstrate their utility in practical supervised learning tasks. Additional details, including theoretical proofs, extended experiments, and in-depth analysis of GMUs, are provided in the supplementary materials, along with code for reproducibility.

**GMU Layer Choice:** In all our experiments, GMUs are placed in the first hidden layer. This design choice is principled: GMUs are a natural extension of radial basis units, which in RBF networks are almost always restricted to a single hidden layer (Buhmann, 2000; Park and Sandberg, 1991; Wurzberger and Schwenker, 2024). Stacking RBF layers is not standard practice, as it produces nested exponentials that risk vanishing or exploding gradients (Goodfellow et al., 2016). Moreover, only recently there has been some work on extending RBFs to multiple layers for deep networks (Wurzberger and Schwenker, 2024), but it requires specialized learning schemes and represents a distinct research direction. In the same spirit, restricting GMUs to the first hidden layer ensures methodological consistency across our experiments. Extending GMUs to deeper architectures is an interesting avenue for future work, but beyond the scope of this foundational study.

### 4.1 SYNTHETIC DATASETS

We design six synthetic sampling scenarios to evaluate GMUs under controlled conditions, each highlighting a different structural or functional property of the data. These experiments test not only whether networks can exploit class-specific structural causal models (SCMs) and generalize to out-of-distribution shifts, but also whether they can capture functional and geometric structure such as Fourier expansions, $k$-subspace thresholds, polynomial boundaries, and Gaussian mixtures. Due to space constraints, we report in the main paper only the *Sparse Linear Structure Prediction* experiment, which directly probes the ability of GMUs to recover sparse latent causes that determine both inputs and labels. The remaining experiments, including Fourier series recovery, $k$-subspace thresholding, sparse neural structure sampling, tree-structured sampling, polynomial boundary sampling, and conditional Gaussian sampling, are presented in the Appendix with full details and results.

#### 4.1.1 SPARSE LINEAR STRUCTURE PREDICTION

**Problem Outline.** We argue that in many natural data sources, only a sparse subset of latent causes is active in generating any given observation. This intuition is consistent with prior work in sparse representation learning (Lee et al., 2006), where sparsity is viewed as a key inductive bias for capturing real-world structure. To test whether GMUs can leverage this property, we construct a synthetic scenario in which the set of active latent causes not only generates the observed input but also directly determines the class label. Concretely, this sampling process is a special case of the CL-SCM framework (Figure 2), where the class-wise models $SCM_1, SCM_2, \ldots, SCM_M$ may share common latent causes and weight structure. The generative process is formally described in Algorithm 1.

**Experimental Setup.** We generate datasets by first sampling the generative weights $W$ from a standard normal distribution $\mathcal{N}(0, 1)$. For each instance, the number of active causes $N_y$ is drawn uniformly from $[1, N_{\max}]$, and the corresponding latent variables are activated to produce the observed input vector $x$. Unless otherwise specified, Gaussian noise with variance $\sigma^2$ is added to each dimension. To probe generalization, we vary the range of the latent variables $\theta_{l_y(i)}$ at test time, thereby creating OOD conditions where the test distribution differs from training. This allows us

| Setup | | | | | Test Config | GMU(1) | GMU(2) | GMU(3) | GMU(4) | GMU(5) | GMU(6) | MLP | MLP (norm.) | R-[1,1] | R-[1,1] (norm.) |
|---|---|---|---|---|---|---|---|---|---|---|---|---|---|---|---|
| $C$ | $N_{total}$ | $d$ | $\sigma$ | $N_{max}$ | | | | | | | | | | | |
| 20 | 25 | 10 | 0.01 | 3 | same | 0.8997 | 0.9311 | 0.9654 | **0.9794** | 0.9694 | 0.966 | 0.9505 | 0.9728 | 0.9622 | 0.9714 |
| 20 | 25 | 10 | 0.01 | 3 | ood | 0.685 | 0.84525 | **0.9735** | 0.8922 | 0.928 | 0.9442 | 0.5725 | 0.7087 | 0.631 | 0.6525 |
| 20 | 25 | 100 | 0.1 | 3 | same | 0.944 | 0.9271 | 0.9348 | **0.9488** | 0.9482 | 0.9482 | 0.9197 | 0.9451 | 0.946 | 0.9462 |
| 20 | 25 | 100 | 0.1 | 3 | ood | 0.8785 | 0.9005 | **0.926** | 0.848 | 0.8592 | 0.9092 | 0.6457 | 0.733 | 0.681 | 0.724 |
| 20 | 25 | 100 | 0.01 | 3 | same | 0.9471 | 0.9494 | 0.9814 | **0.9874** | 0.9862 | 0.9825 | 0.9545 | 0.9754 | 0.9654 | 0.9788 |
| 20 | 25 | 100 | 0.01 | 3 | ood | 0.892 | 0.9215 | **0.981** | 0.9335 | 0.9365 | 0.9612 | 0.6282 | 0.7492 | 0.693 | 0.7522 |
| 20 | 25 | 100 | 0 | 3 | same | 0.9474 | 0.9565 | 0.9962 | 0.9951 | 0.9974 | **0.9991** | 0.9548 | 0.9797 | 0.9622 | 0.9868 |
| 20 | 25 | 100 | 0 | 3 | ood | 0.892 | 0.925 | **0.9977** | 0.9915 | 0.9835 | 0.9905 | 0.6275 | 0.7495 | 0.6842 | 0.7432 |
| 20 | 25 | 500 | 0.01 | 3 | same | 0.9502 | 0.9434 | 0.9831 | 0.9891 | **0.9894** | 0.9874 | 0.9537 | 0.9771 | 0.9685 | 0.9845 |
| 20 | 25 | 500 | 0.01 | 3 | ood | 0.9087 | 0.9215 | **0.9845** | 0.948 | 0.9655 | | 0.59725 | 0.7677 | 0.6467 | 0.7665 |
| 20 | 25 | 1000 | 0.01 | 3 | same | 0.9111 | 0.9502 | 0.9834 | 0.9957 | 0.9914 | **0.996** | 0.9662 | 0.9814 | 0.9762 | 0.9845 |
| 20 | 25 | 1000 | 0.01 | 3 | ood | 0.8657 | 0.9017 | 0.975 | 0.9537 | **0.987** | 0.983 | 0.517 | 0.7525 | 0.6572 | 0.7837 |
| 20 | 25 | 1000 | 0.01 | 6 | same | 0.978 | 0.9385 | 0.9791 | 0.9908 | 0.9885 | **0.9951** | 0.9637 | 0.9882 | 0.9757 | 0.9882 |
| 20 | 25 | 1000 | 0.01 | 6 | ood | 0.9272 | 0.6867 | 0.9462 | 0.9462 | 0.9905 | **0.9997** | 0.6407 | 0.9107 | 0.641 | 0.904 |
| 50 | 10 | 1000 | 0.01 | 3 | ood | 0.5752 | 0.743 | 0.8997 | **0.9077** | 0.891 | 0.8712 | 0.5492 | 0.5575 | 0.4132 | 0.5352 |
| **Wins/Losses/Ties vs GMU(3)** | | | | | | 14/1/0 | 15/0/0 | – | 7/7/1 | 6/9/0 | 6/9/0 | 15/0/0 | 12/3/0 | 14/1/0 | 10/5/0 |
| **Wilcoxon $p$-value** | | | | | | 1.8e-04 | 6.1e-05 | – | 0.432 | 0.639 | 0.804 | 6.1e-05 | 8.4e-03 | 4.3e-04 | 1.8e-02 |

**Table 1:** Test accuracy results on the sparse linear structure prediction experiments.

to evaluate whether models capture the underlying generative mechanism or merely memorize the training data.

**Baselines and Models.** We compare GMUs against strong feedforward baselines. Specifically, we evaluate a two-layer MLP with 512 ReLU-activated hidden units (denoted MLP-512) and a ResNet with two groups, each containing one block of 512 units (denoted ResNet-512-[1,1]). To ensure fairness, all models are trained with the same optimizer and learning rate schedule. We also evaluate normalized variants of the baselines, where inputs are scale-normalized and mean-adjusted, denoted with the suffix "(norm.)". For GMUs, we use a single GMU layer with $d$ inputs and $C$ outputs, denoted GMU($k$), where $k$ indicates the order of the GMU. This design allows us to directly test whether the inductive bias of GMUs provides an advantage over standard architectures.

**Results.** The results are summarized in Table 1. GMUs consistently outperform both MLP and ResNet baselines across parameter settings, with the gap most pronounced in OOD scenarios where test distributions differ from training. This suggests that GMUs capture aspects of the underlying generative process better than the other counterparts. We also find that higher-order GMUs ($k > N_{\max}$) perform comparably to $k = N_{\max}$, since larger-order GMUs can emulate lower-order ones by simply setting some weights to zero. The Wilcoxon signed-rank test shows no significant gains once $k$ exceeds the true number of active causes, indicating that while overparameterization is not harmful, its benefits saturate beyond the underlying generative complexity.

**Discussions.** These findings highlight two key points. First, GMUs are well-suited for tasks where the discriminative signal is tied to a sparse generative structure, a property that is common in many real-world domains (e.g., vision, genomics, and language). Second, the robustness of GMUs under OOD shifts suggests that their inductive bias aligns more closely with the true data-generating process in this scenario. In the Appendix, we extend this analysis to five other synthetic scenarios listed above, which probe complementary aspects of GMU behavior such as functional approximation (Fourier series), geometric discrimination ($k$-subspace thresholding), non-linear causal sampling (sparse neural structure), hierarchical observed-variable dependencies (tree-structured sampling), and non-linear decision boundaries (polynomial and Gaussian mixtures). Together, these experiments provide a broad and rigorous evaluation of GMUs under diverse synthetic conditions.

### 4.2 TABULAR DATASETS (REAL)

**Outline:** We test and compare performance on 27 OpenML datasets, selecting a diverse subset from Kadra et al. (2021), ranging from small-scale (800 samples, 5 dimensions) to medium-scale (170k samples, 1k dimensions). For the Resnet-512-[1,1] architecture, which performed well in our synthetic experiments overall, we replace the first layer with four types of GMU units: $k = 0, 1, 2, 3$. Each configuration maintains 128 units in the first layer, yielding 512 output units, matching the Resnet. We denote this network as GMU-Resnet-512-[1,1] (or GMU-S). Since GMU-Resnets are slightly more parameterized than Resnets, we scaled the Resnets (530 units per layer) to ensure equal parameter counts. Results are provided in Table 3.

**Takeaways:** We find that overall, in 23 out of 27 cases, GMU-Resnet-512-[1,1] showcases better or on-par balanced accuracy. Furthermore, the GMU-Resnet-512-[1,1] performs competitively or favorably against other state-of-the-art approaches in Kadra et al. (2021) when compared one-to-one via the Wins/Losses/Ties criterion (please see supplementary materials).

| Models | Datasets | | | | | | | | | | | | | |
|---|---|---|---|---|---|---|---|---|---|---|---|---|---|---|
| | credit-g | anneal | kr-vs-kp | mfeat | vehicle | kc1 | phoneme | cnae-9 | blood | Australian | car | segment | jasmine | sylvine |
| R | 0.6972 | 0.8546 | **0.9969** | **0.9850** | 0.8446 | **0.7104** | **0.8959** | **0.9398** | 0.6304 | 0.8479 | **1.0000** | 0.9254 | 0.7438 | 0.9250 |
| GMU-R | **0.7036** | **0.8610** | 0.9969 | 0.9800 | **0.8793** | 0.6866 | 0.8882 | 0.9398 | **0.6718** | **0.8726** | 1.0000 | **0.9307** | **0.7520** | **0.9268** |
| | adult | nomao | bank | jungle | volkert | helena | connect-4 | higgs | numerai | walking | ldpa | aloi | skin-seg | |
| R | 0.7652 | 0.9582 | 0.7261 | 0.9686 | 0.6766 | **0.2213** | 0.7369 | 0.6684 | 0.5066 | 0.6203 | **0.6991** | 0.9617 | **0.9997** | |
| GMU-R | **0.7735** | **0.9599** | **0.7382** | **0.9773** | **0.7003** | 0.2206 | **0.7535** | **0.6781** | **0.5146** | **0.6322** | 0.6777 | **0.9684** | 0.9996 | |

**Table 3:** Balanced Accuracy on 27 Tabular datasets from OpenML.

| | Dataset: MNIST | | | | | | | | | | | | | Dataset: smallNORB | | |
|---|---|---|---|---|---|---|---|---|---|---|---|---|---|---|---|---|
| Network | Standard | brightness | canny | dotted | fog | glass | impulse | motion | shot | spatter | zigzag | Average | | Model | Test Accuracy (Uncorrupted) | Test Accuracy (Corrupted) |
| CNN | 0.9949 | 0.2274 | 0.6149 | 0.9791 | 0.1188 | 0.541 | 0.4529 | **0.9675** | 0.9226 | **0.9834** | 0.7826 | 0.6895 | | | | |
| GMU-CNN | 0.9954 | **0.9913** | **0.8998** | **0.9896** | **0.9234** | **0.8141** | **0.9377** | 0.9614 | **0.9449** | 0.9759 | **0.9447** | **0.9434** | | CNN | 0.954321 | 0.436872 |
| | Dataset: Fashion-MNIST | | | | | | | | | | | | | GMU-CNN (1S-1D) | **0.97477** | 0.546502 |
| CNN | 0.9329 | 0.4535 | 0.3709 | 0.8786 | 0.2712 | **0.6518** | 0.2056 | **0.7188** | **0.5959** | 0.8835 | 0.8131 | 0.615982 | | GMU-CNN (2D-1S) | 0.96642 | 0.516132 |
| GMU-CNN | 0.9356 | **0.8250** | **0.7058** | **0.9118** | **0.7442** | 0.5817 | **0.6703** | 0.6831 | 0.48964 | **0.8868** | **0.8806** | **0.755867** | | GMU-CNN (3S-1D) | 0.96 | **0.56329** |

|  (a)  |  (b)  |
|---|---|

| | Dataset: CIFAR-10 | | | | | | | | | | | | | | |
|---|---|---|---|---|---|---|---|---|---|---|---|---|---|---|---|
| Network | Standard | brightn | contrast | defocus | elastic | fog | gauss_blur | glass | impulse | motion | pixelate | saturate | shot_noise | spatter | Average |
| VGG-16 | 0.8594 | 0.83642 | 0.5952 | 0.68148 | 0.71436 | 0.74422 | 0.5929 | 0.44146 | 0.59338 | 0.6211 | 0.6826 | 0.8147 | 0.6366 | 0.7440 | 0.669109 |
| GMU(3)-VGG | **0.8645** | 0.8540 | 0.7489 | 0.7811 | 0.7493 | 0.8176 | 0.741 | **0.4665** | **0.6003** | 0.7142 | 0.7192 | 0.8211 | **0.6372** | 0.7861 | 0.725885 |
| GMU(8)-VGG | **0.8734** | **0.86188** | **0.75478** | **0.7977** | **0.7534** | **0.8311** | **0.7620** | 0.4383 | 0.5608 | **0.7325** | **0.7273** | **0.8298** | 0.6163 | **0.7874** | **0.727174** |

(c)

**Table 4:** Test Accuracy on standard and corrupted data on (a) MNIST and Fashion-MNIST (b) smallNORB, and (c) CIFAR-10 datasets.

### 4.3 VISION DATASETS (REAL)

**Outline:** We construct convolutional GMUs, where we replace every linear operation in convolution with a GMU, using which we create GMU-CNN architectures, where the first layer is replaced with only convolutional GMUs and the rest of the network is unchanged. We focus on two aspects: whether the GMUs show better generalization compared to CNNs with the same architecture and whether they show better out-of-distribution performance to test-time corrupions. We report our findings across four datasets: MNIST, Fashion-MNIST, CIFAR-10, smallNORB and their corrupted versions. Results are summarized in Table 4. Additionally, to also showcase that GMUs can be combined with benchmark approaches for out-of-distribution settings such as test time adaptation (You et al., 2021; Lim et al., 2023), we provide additional experiments in the supplementary materials.

**Takeaways:** We find that GMU-CNNs show substantial improvements in terms of robustness to test-time corruptions in all cases. In all cases, we see that GMU-CNNs show improvements when compared with their native CNN baselines. On smallNORB, GMU-CNNs reach performance competitive to other benchmarks such as first generation capsule networks (2.5 vs 2.7% error rate in Patrick et al. (2022)). We don't use data-augmentation or any other regularization approach in these experiments, to focus on the intrinsic generalization performance.

**Computation times:** We find that GMU-CNNs and GMU-MLPs still maintain instantaneous inference times per sample, in spite of inverse operations. This is because the size of the matrix being inverted in equation 3 is $k \times k$, where $k$ is the order of the GMU. In all our experiments $k \leq 8$, which makes this operation efficient. Overall, in the tabular data experiments, the average per-sample inference time of GMU-Resnets was 0.017 ms compared to 0.004 ms for Resnets, and for the vision experiments, GMU-CNNs took 0.0265 ms compared to 0.0083 ms for CNNs.

### 4.4 CROSS-DATASET TESTING (REAL)

In this section, we train GMU-ResNets and ResNets (with the ResNet-18 backbone) on the Street View House Numbers dataset (SVHN, Netzer et al. (2011)) and then test directly on MNIST, without any changes to the network. Similarly, we use

**Table 2:** Cross-Dataset Testing

| Test Setting | GMU | Standard |
|---|---|---|
| SVHN Only | **96.58** | 95.60 |
| SVHN → MNIST | **83.99** | 67.28 |
| MNIST → SVHN | **54.50** | 19.95 |

trained GMU-CNNs and CNNs on MNIST and test on SVHN. The results are shown in Table 2. We not only see GMU-Resnet-18 significantly improve over Resnet-18 on native SVHN, but we also see significant gains in accuracy when directly transferring between datasets.

## 5 CONCLUSION AND LIMITATIONS

Our work demonstrates the potential advantages of an alternative computational unit that computes from a generative perspective, imposing a low-complexity constraint on the generation process. Generative Matching Units showcase better generalization and demonstrate a significantly higher ability to identify dynamic causal structures in the inputs. On real tabular datasets, Resnets replaced with GMU layers in the first layer show signifcant performance improvements. Many possibilities

remain open for incorporating GMUs in larger networks across other domains, and also finding ways to cascade multiple GMU layers.

**Limitations:** Our experiments focus on GMUs replacing traditional units only in the first layer. This choice is driven by two factors: First, replacing the first layer already leads to significantly different network behavior, making it a natural point of study. Second, while GMUs introduce negligible overhead in the first layer, computational costs increase in deeper layers. We are actively working on optimizing efficiency to scale GMUs to ImageNet-level datasets without added compute burden.

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

APPENDIX TABLE OF CONTENTS

## A    SUMMARY OF SUPPLEMENTARY MATERIALS

In this document we share additional experiments and analyses, empirical details and theoretical proofs and any further clarifications. Code is provided here, which includes a GUI based demonstration of the performance of trained GMU-CNNs for digit classification and Fashion item classification. Instructions for the demo are available here.

## B VISION: COMPARING WITH TEST-TIME ADAPTATION

| Corruption Type | Standard (CNN) | TTA (CNN) | Standard (GMU-CNN) | TTA (GMU-CNN) |
|---|---|---|---|---|
| Brightness | 0.2274 | 0.9887 | 0.9913 | **0.9933** |
| Canny Edges | 0.6149 | 0.9431 | 0.8998 | **0.9813** |
| Dotted Line | 0.9791 | 0.9873 | 0.9896 | **0.9902** |
| Fog | 0.1188 | 0.6074 | 0.9234 | **0.9609** |
| Impulse Noise | 0.4529 | 0.9098 | 0.9377 | **0.9683** |
| Motion Blur | 0.9675 | **0.9858** | 0.9614 | **0.9666** |
| Shot Noise | 0.9226 | 0.9677 | 0.9449 | 0.9757 |
| Spatter | 0.9834 | **0.9857** | 0.9759 | 0.9755 |
| Zigzag | 0.7826 | 0.8707 | 0.9447 | **0.9608** |
| Average | 0.67213333 | 0.916244 | 0.952078 | **0.974733** |

**Table 5:** Accuracy comparison of standard (no TTA) and Test-Time Adaptation (TTA) approaches across various corruption types on the MNIST dataset.

| Corruption Type | Standard (CNN) | TTA (CNN) | Standard (GMU-CNN) | TTA (GMU-CNN) |
|---|---|---|---|---|
| Brightness | 0.4535 | 0.8281 | 0.8250 | **0.8550** |
| Canny Edges | 0.3709 | 0.5322 | 0.7058 | **0.7575** |
| Dotted Line | 0.8786 | 0.8953 | 0.9118 | **0.9161** |
| Fog | 0.2712 | 0.6060 | 0.7442 | **0.8373** |
| Impulse Noise | 0.2056 | 0.6614 | 0.6703 | **0.7837** |
| Motion Blur | 0.7188 | **0.7865** | 0.6831 | 0.7830 |
| Shot Noise | 0.5959 | **0.7059** | 0.4896 | 0.6870 |
| Spatter | 0.8835 | 0.8902 | 0.8868 | **0.8908** |
| Zigzag | 0.8131 | 0.8606 | 0.8806 | **0.8941** |
| Average | 0.6159 | 0.7518 | 0.7558 | **0.8227** |

**Table 6:** Accuracy comparison of standard (no TTA) and Test-Time Adaptation (TTA) approaches across various corruption types on the Fashion-MNIST dataset.

| Corruption Type | Standard (VGG) | TTA (VGG) | Standard (GMU(3)) | TTA (GMU(3)) | Standard (GMU(8)) | TTA (GMU(8)) |
|---|---|---|---|---|---|---|
| Brightness | 0.83642 | 0.82376 | 0.8540 | 0.85266 | 0.86188 | **0.86362** |
| Contrast | 0.5952 | 0.74014 | 0.7489 | 0.8016 | 0.75478 | **0.81086** |
| Defocus Blur | 0.68148 | 0.79334 | 0.7811 | 0.82878 | 0.7977 | **0.83792** |
| Elastic Transform | 0.71436 | 0.74232 | 0.7493 | 0.7808 | 0.7534 | **0.78318** |
| Fog | 0.74422 | 0.77798 | 0.8176 | 0.8283 | 0.8311 | **0.8393** |
| Frost | 0.7386 | 0.7386 | 0.7410 | 0.77508 | 0.7620 | **0.78332** |
| Gaussian Blur | 0.5929 | 0.77528 | 0.7410 | 0.81844 | 0.7620 | **0.82662** |
| Impulse Noise | 0.59338 | 0.65226 | 0.6003 | **0.75382** | 0.5608 | 0.73398 |
| Motion Blur | 0.6211 | 0.74056 | 0.7142 | 0.79238 | 0.7325 | **0.79896** |
| Pixelate | 0.6826 | 0.7591 | 0.7192 | 0.80666 | 0.7273 | **0.81328** |
| Saturate | 0.8147 | 0.81246 | 0.8211 | 0.8283 | 0.8298 | **0.83882** |
| Shot Noise | 0.6366 | 0.72318 | 0.6372 | **0.77136** | 0.6163 | 0.76632 |
| Spatter | 0.7440 | 0.73732 | 0.7861 | 0.79432 | 0.7874 | **0.7989** |
| Average | 0.691966 | 0.7551 | 0.747 | 0.8025 | 0.752074 | **0.807314** |

**Table 7:** Accuracy comparison of standard (no TTA) and Test-Time Adaptation (TTA) approaches across various corruption types on the CIFAR-10 dataset.

We extend our analysis to include test-time adaptation (TTA) and summarize the statistical comparisons in Table 8. The table reports wins, losses, and ties across corruption types, along with Wilcoxon signed-rank $p$-values for GMU-CNN versus standard CNN (or VGG in the case of CIFAR-10), both under standard evaluation and with TTA. Several clear trends emerge. First, under standard evaluation, GMU-CNN significantly outperforms the baseline on MNIST and CIFAR-10, and shows a positive but not statistically significant trend on Fashion-MNIST. Second, when TTA is applied, both standard and GMU-based models benefit substantially, confirming that adaptive batch normalization improves robustness to distribution shifts. Crucially, GMU-based architectures continue to hold an advantage even after TTA is applied to both models: on Fashion-MNIST and CIFAR-10, the improvements are statistically significant ($p = 0.01953$ and $p = 0.00012$, respectively), while on MNIST the trend remains favorable though not significant ($p = 0.06445$). Finally, higher-order GMU variants (e.g., GMU(8) on CIFAR-10) show that the benefits of TTA and GMU are complementary, leading to consistent wins across all corruptions. These results demonstrate that GMU-based architectures not only integrate seamlessly with existing TTA methods but also preserve their relative advantage over conventional networks under test-time distribution shifts.

| Dataset | Standard | | | With TTA | | |
|---|---|---|---|---|---|---|
| | Wins | Losses | Wilcoxon $p$ | Wins | Losses | Wilcoxon $p$ |
| MNIST | 9 | 2 | 0.00488 | 7 | 2 | 0.06445 |
| Fashion-MNIST | 8 | 3 | 0.08740 | 7 | 2 | 0.01953 |
| CIFAR-10 | 11 | 3 | 0.00336 | 13 | 0 | 0.00012 |

**Table 8:** Comparison of GMU-CNN vs CNN across datasets. Left: standard evaluation. Right: test-time adaptation (TTA). Wins/losses are counted per corruption type; $p$-values are from the one-sided Wilcoxon signed-rank test (GMU-CNN > CNN).

# C  ADDITIONAL EXPERIMENTS

In this section, we demonstrate the versatility of GMUs in achieving good generalization performance in diverse synthetic settings. We include the following experimental setups:

1. **Fourier series function v/s Noise Differentiation**: We show how single-layered GMU architectures can learn underlying function structures in the data, when the functions are represented generally as a Fourier series.

2. $k$**-subspace distance based thresholding**: We show that the problem that is the most natural test for a single-layered GMU architecture, $k$-subspace distance based thresholding, is significantly harder for other networks.

3. **Sparse Neural Structure Sampling:** Linear GMUs are motivated from a linear structural causal model's sampling process, as observed in Section 2. So, it is natural to ask whether the linear structure assumption in linear GMUs can still impart any benefits when the underlying data is generated non-linearly. Specifically, following the CL-SCM sampling as defined in Section 2, the generating function $f$ in the SCM of Figure 1 (c) is set to a 2-layered neural network.

4. **Tree-structured sampling:** We evaluate GMUs on data generated via a synthetic SCM in which the generating causes are the observed input dimensions themselves; each class is associated with a distinct tree over the observed variables, and samples are generated by propagating from a root variable with additive noise.

5. **Polynomial Boundary Sampling**: Here, we consider the underlying ground truth distribution to be separable by a non-linear, polynomial decision boundary. Linear decision boundaries are not realistic for most natural datasets, and thus this provides a natural scalability test for GMUs.

6. **Conditional Gaussian Sampling:** This is a well-known and studied setting where the conditional distributions $P(X|Y = i)$ are all Gaussian. We vary the overlap between the Gaussians and test whether GMUs can robustly learn in this setting.

## C.1  FOURIER SERIES FUNCTION V/S NOISE DIFFERENTIATION

**Problem Setup:** We consider a scenario where $x \in \mathbb{R}^d$ can belong to two categories: fourier series function or noise. In the noise category, $x$ is generated as $x \sim \mathcal{U}(-0.5, 0.5)^d$, where $\mathcal{U}(a, b)$ denotes the uniform distribution sampled from the range $(a, b)$. In the fourier series function category, the $i^{th}$ dimension of $x$, $x_i$ is generated as $x_i = a_0 + \sum_{j=1}^{K} a_j sin(10 * i/d) + b_j cos(10 * i/d)$. Here $a_0, a_1, ..a_K$ and $b_1, b_2, .., b_K$ are coefficients of the fourier series representation of $X$. Thus, when $x$ is a fourier series function, the dimensions of $x$ are essentially sampled from the outputs of a fourier series representation. We consider $K = 2$ for our experiments. The objective of this experiment is to identify if $x$ is a noisy signal, or has a function pattern via a fourier series representation.

**GMU construction:** We design a GMU as follows.

$$\text{gmu}(x, w) = \exp\left(-\min_\theta \mathbb{E}_i\left[(f(\theta, w_i) - x_i)^2\right]\right),$$

where $f(\theta, w_i) = \theta_0 + \sum_{j=1}^{k} (w_i)^j \theta_j$. This GMU, in a nutshell, can find to what degree the input dimensions of $x = [x_1, x_2, .., x_d]$ can be represented as a polynomial function $x_i = f(\theta, w_i)$ of the weights.

**Results:** We compare the performance of a single GMU-k unit, where $k$ is its order, with a single perceptron unit, a 2-layer $\mu$-RBF networks having 512 $\mu$-RBF units in the first layer [1], a relu-activated

---
[1] Note that we tried training with the additional $\sigma$ term, but we got best results without it.

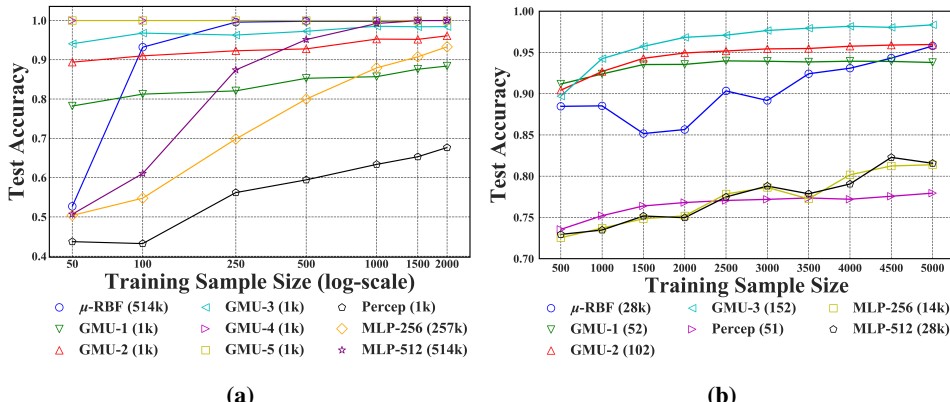

**(a)**                                   **(b)**

**Figure 4:** Comparing test accuracy of a single GMU with other approaches on a Fourier series function v/s noise differentiation and b $k$-subspace distance based thresholding experiments. The parameter counts for each model are given within the brackets.

single hidden layer neural network with 256 units (MLP-256) and with 512 units (MLP-512). The results are shown in Figure 4a. We also report the parameter counts for comparison. Overall, we find that GMU-4 and GMU-5 quickly capture the underlying structure, achieving 100% test accuracy with only 50 samples. In contrast, we see that the MLP and RBF variants need more data and parameters to achieve similar performance.

## C.2 $k$-SUBSPACE DISTANCE BASED THRESHOLDING

**Problem Setup:** Just as a single-layer perceptron aligns with problems where the underlying distribution is linearly separable, and similarly, a single RBF neuron can perfectly classify data when the distribution is separable by a sphere, GMUs address an equivalent problem in a different geometric sense. We consider a supervised classification problem where the datapoint $x \in \mathbb{R}^d$ is categorized according to the distance it has to a $k$-dimensional subspace (i.e., $k$-subspace distance in Section 3.3), represented by the basis vectors as $U = Span\{u_1, u_2, .., u_k\}$, where $u_i \in \mathbb{R}^d$. Let the distance between $U$ and $x$ be denoted by $d(x, U)$. Then, given a threshold distance $d_{thres}$, we categorize $x$ as the category 0 if $d(x, U) \leq d_{thres}$ and the category 1 if otherwise. To construct the training data, we sample $x$ from the uniform distribution $x \sim \mathcal{U}(0, 1)^d$ and each element of $u_1, u_2, .., u_k$ is also chosen randomly from $\mathcal{U}(0, 1)$. Then, $d_{thres}$ is chosen such the number of samples per class are same for both categories. We use single-layer linear GMU networks for classification. In this case, the shape of the decision boundary exactly aligns with thresholding the single GMU output.

**Results:** We compare the performance of a single linear GMU-$k$ unit, with a single perceptron unit, 2-layer $\mu$-RBF networks having 512 $\mu$-RBF units in the first layer, a relu-activated single hidden layer neural network with 256 units (MLP-256) and another one with 512 units (MLP-512). The results are shown in Figure 4b. Unsurprisingly, we find that GMU-3 significantly outperforms all other approaches for all sample sizes. Interestingly, although we see the $\mu$-RBF catching up to the GMUs performance with more data, but the MLP-variants still show significant drop in terms of accuracy, demonstrating the hardness of this function fitting task from the perspective of MLPs.

**Takeaways:** These results highlight the hardness of the task for traditional architectures. While GMU-3 consistently achieves high classification accuracy, MLP variants struggle to generalize, exhibiting a significant accuracy drop even with more data. Overfitting remains a concern, particularly for deeper networks, as they fail to capture the structured geometric separability essential for robust generalization.

## C.3 SPARSE NEURAL STRUCTURE SAMPLING

**Problem Outline:** We consider the case when the underlying data follows a CL-SCM sampling process (Figure 1 (b)). However, unlike the sparse linear structure experiments in the main paper where the generating function $f$ of the SCMs is linear, we consider non-linear $f$. In particular, we consider the case when $f$ is a two layered neural network with tanh activations in each layer. Following the CL-SCM structure, each class has its own neural network weights it uses to generate the input. We formally outline the sampling process in Algorithm 2.

**Algorithm 2** Sparse Neural Structure Sampling

---

1: **Setup:** $X \in \mathbb{R}^d$, labels $y \in \{1, \ldots, C\}$, class-specific weights $W_y^{(1)} \in \mathbb{R}^{N_{\text{total}} \times h}$, $W_y^{(2)} \in \mathbb{R}^{h \times d}$, latent variables $\theta \in \mathbb{R}^{N_{\text{total}}}$
2: **Latent Variable Sampling:** $\theta_i \sim \mathcal{U}(0,1)$ for all $i \in \{1, \ldots, N_{\text{total}}\}$
3: **Generative Process:**
4:     Sample $y \sim \text{Unif}\{1, \ldots, C\}$
5:     Compute $x$:
$$x = \tanh\left(W_y^{(2)} \tanh\left(W_y^{(1)} \theta\right)\right) + \epsilon$$
6: where $\epsilon \sim \mathcal{N}(0, \sigma^2 I_d)$

---

**Experiments:** Here, we focus on testing and comparing the scalability of GMUs with MLPs, RBFs and Resnet baselines, in two different settings. For each setting we consider two layered neural network SCMs with 32 hidden units ($h = 32$ in Algorithm 2), $\sigma = 0.01$, $d = 10$ and $C = 20$. In the first setting, we consider a total of 2 latent generating variables (i.e., $N_{total} = 2$ in Algorithm 2) and consider $N_{total} = 5$ for the second setting.

To rigorously assess whether the networks are truly learning the underlying function, we introduce a structured distribution shift: during training, latent variables $\theta_i$ are sampled from $\mathcal{U}(0, 0.5)$ or $\mathcal{U}(0.5, 1.0)$, while in testing, they are drawn from the opposite interval. This avoids any data overlap and thus prevents memorization as a reliable strategy to improve performance. In each setting, we plot the test accuracy trends in response to dataset size, and also plot the test accuracy trends of all networks w.r.t their complexity (parameter count). For the dataset size trends, each network's number of hidden neurons per layer was fixed to 256. The results are shown in Figure 5.

**Observations:** We find that GMU-MLPs show better scaling to data size in this scenario, when compared to the other baselines. Furthermore, we find that higher order GMUs showcase better performance overall. Interestingly, we see that the response to data size increase can even be negative for MLPs (e.g. when $N_{total} = 5$), which highlights the hardness of the generalization problem in this case. When considering the accuracy versus model complexity plots, we see that GMU architectures are closest to the pareto front, although there is a visible decrease in accuracy that accompanies GMU architectures when the parameter count increases beyond a point. Interestingly, the results hint at an optimal range of parameter count for the GMU architectures to significantly outperform the baselines.

## C.4  TREE-STRUCTURED SAMPLING

**Problem Setup.** This synthetic experiment is a classification benchmark designed to probe GMUs under structured dependencies in which *observed variables themselves act as causal nodes*. Each class $y$ is associated with a distinct directed spanning tree $G_y$ defined over the $d$ observed input coordinates in $X = [x_1, \ldots, x_d]$. Samples are generated by propagating a value sampled at the class-specific root down the directed edges with additive edge noise (Algorithm 3). This setting contrasts with latent-only CL-SCM setups because the causal graph is explicit over input dimensions; consequently the sampling process induces hierarchical linear relations among observed features that we expect GMUs to exploit. Unless stated otherwise we fix the root variance $\sigma_0 = 0.1$ and set the nominal edge coefficients $\alpha_i = 1$.

**Tree construction and branching factor.** Each class tree $G_c$ is constructed at the start of a run by selecting a root uniformly from $\{1, \ldots, d\}$ and growing a directed spanning tree in a breadth-first fashion. The branching factor $\Delta$ controls how many children a node may receive during construction: larger $\Delta$ yields shallower, wider trees while smaller $\Delta$ yields deeper, narrower trees. We record the parent map $pa(\cdot, G_c)$ for each class and fix the tree topology per seed. Specifying $\Delta$ alongside $d$ (as in Table 10) removes ambiguity about tree shape and clarifies how path length and noise accumulation vary across experiments.

**Generative process and configurations.** A datum is generated by sampling a class label $y \sim \text{Unif}\{1, \ldots, N_y\}$, sampling the root value $x_{r(G_y)} \sim \mathcal{N}(0, \sigma_0^2)$, and then for each non-root node $i$ in topological order setting

$$x_i \leftarrow \alpha_i \, x_{pa(i, G_y)} + \epsilon_i,$$

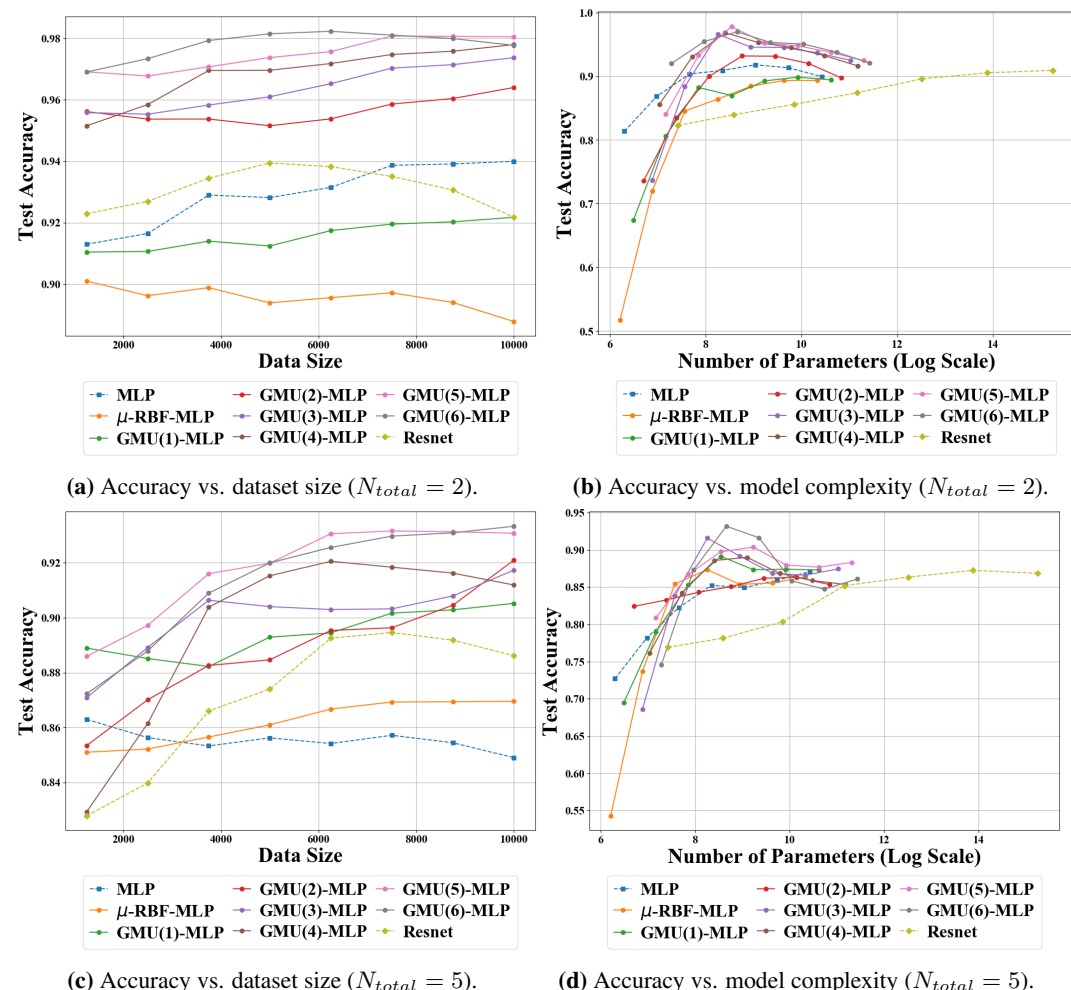

**(a)** Accuracy vs. dataset size ($N_{total} = 2$).      **(b)** Accuracy vs. model complexity ($N_{total} = 2$).

**(c)** Accuracy vs. dataset size ($N_{total} = 5$).      **(d)** Accuracy vs. model complexity ($N_{total} = 5$).

**Figure 5:** Scalability studies in two experimental settings in the class-wise sparse neural structure sampling experiments. The underlying SCM here is a 2-layered tanh-activated neural network. We report accuracy trends in response to training dataset size changes in (a) and (c), and in response to parameter count changes in (b) and (d). Please note that only the MLP and Resnet baselines are represented via dashed lines.

with $\epsilon_i \sim P_\epsilon$. The experimental difficulty and the types of distributional shifts we evaluate are controlled via the choice of $P_\epsilon$ and whether the edge coefficients $\alpha_i$ are fixed or resampled. Table X (below) lists the abbreviations used throughout (G, GS, GM1, GM2, MG, MGN) and their precise sampling rules; for example, **G** denotes Gaussian edge noise with $\epsilon_i \sim \mathcal{N}(0, 0.1)$, while **GM1** and **GM2** introduce multiplicative scaling by resampling $\alpha_i$ at each generative pass from broader Normal distributions (see table for exact parameters). All shift configurations preserve the tree topology while changing effective coupling strengths or variances along edges, producing class-conditional but distributionally shifted test data.

**Train/Test nomenclature and reproducibility.** The Train/Test columns in Table 10 indicate the generative configuration used for the corresponding split (for instance, Train = G, Test = GM1). When an experiment fixes $\alpha_i$ during training but resamples or perturbs it at test time we indicate this using the MG/MGN abbreviations described in the table. Trees $\{G_1, \ldots, G_{N_y}\}$ are resampled for each experimental seed and then held fixed within that seed to ensure repeatability of class-conditional structure across samples.

**Experimental protocol.** For the results reported in Table 10 we fix $N_y = 10$, $\sigma_0 = 0.1$, and generate 500 training and 3500 test samples per configuration. We compare GMU($k$)-MLP-512 (a single GMU layer whose outputs feed a 512-unit linear head), MLP-512, ResNet-512-[2,1], and ResNet-512-[2,2]. All models are trained discriminatively using supervised cross-entropy; GMUs are not trained with

| Abbrv. | Description |
|---|---|
| G | Gaussian: $\epsilon_i \sim \mathcal{N}(0, 0.1)$ |
| GS | Skewed Gaussian: $\epsilon_i \sim 0.1 \cdot \text{SN}(0, 1, 4)$ |
| GM1 | Gaussian with multiplicative scaling: $\epsilon_i \sim \mathcal{N}(0, 0.1)$ and $\alpha_i \sim \mathcal{N}(0, 1)$ (randomly sampled in each generative pass) |
| GM2 | Gaussian with multiplicative scaling: $\epsilon_i \sim \mathcal{N}(0, 0.1)$ and $\alpha_i \sim \mathcal{N}(0, 4)$ (randomly sampled in each generative pass) |
| MG | Gaussian with fixed multiplicative scaling: $\epsilon_i \sim \mathcal{N}(0, 0.1)$, $\alpha_i \sim \mathcal{N}(0, 1)$ (Fixed) |
| MGN | Gaussian with noise-added multiplicative scaling: $\epsilon_i \sim \mathcal{N}(0, 0.1)$, $\alpha_i \sim \alpha_i^{\text{train}} + \mathcal{N}(0, 1)$ where $\alpha_i^{\text{train}}$ is the $\alpha_i$ fixed at training time |

**Table 9:** Table of Abbreviations for the Dynamic tree prediction experiments.

---

**Algorithm 3** Tree-structured sampling (SCM over observed variables)

---

1: **Setup:** input dimension $d$, number of classes $C$, branching factor $\Delta \in \mathbb{N}$, edge coefficients $\alpha \in \mathbb{R}^d$, root variance $\sigma_0^2$, edge-noise distribution $P_\epsilon$
2: **Tree construction (per class $c$):**
3:     Pick a root node $r_c$ uniformly from $\{1, \ldots, d\}$
4:     Initialize frontier $F \leftarrow \{r_c\}$, visited $V_c \leftarrow \{r_c\}$
5: **while** $|V_c| < d$ **do**
6:     Pop node $u$ from $F$
7:     Sample $b \sim \min(\Delta, d - |V_c|)$ (number of children for $u$)
8:     Select $b$ nodes uniformly at random from $\{1, \ldots, d\} \setminus V_c$ and add directed edges $(u \to v)$ to $E_c$
9:     Add selected nodes to $V_c$ and append them to $F$
10: **end while**
11: Ensure $G_c = (V_c, E_c)$ is a directed spanning tree over $\{1, \ldots, d\}$
12: **Generative process (sample one datapoint):**
13:     Sample class label $y \sim \text{Unif}\{1, \ldots, C\}$
14:     Sample root value $x_{r_y} \sim \mathcal{N}(0, \sigma_0^2)$
15: **for** each non-root node $i$ in topological order of $G_y$ **do**
16:     Sample noise $\epsilon_i \sim P_\epsilon$
17:     Set $x_i \leftarrow \alpha_i x_{pa(i, G_y)} + \epsilon_i$
18: **end for**
19: **Return:** $(X = [x_1, \ldots, x_d], y)$

---

any reconstruction objective in these experiments. Reported accuracies are averages over random seeds (tree topology, noise draws, and any sampled $\alpha_i$ variations) and the table explicitly lists the branching factor $\Delta$ used for each run.

**Why $\Delta$, tree shape and shift types matter.** The branching factor and resulting tree geometry determine path lengths over which signal and noise accumulate: deeper trees propagate root information through more multiplicative steps, making downstream coordinates more sensitive to edge-wise scaling and noise; wider trees distribute signal across many shallow branches, changing how discriminative information is localized. The different abbreviations (G, GS, GM1, GM2, MG, MGN) correspond to structured perturbations that either alter noise skew/scale or resample multiplicative edge coefficients at train or test time (see the abbreviations table). These controlled variations let

| Setup ($N_y = 10$, $\sigma_0 = 0.1$) | | | | GMU(0)-MLP | GMU(1)-MLP | GMU(2)-MLP | GMU(3)-MLP | GMU(4)-MLP | GMU(5)-MLP | GMU(6)-MLP | GMU(7)-MLP | GMU(8)-MLP | MLP | R-[2,1] | R-[2,2] |
|---|---|---|---|---|---|---|---|---|---|---|---|---|---|---|---|
| $d$ | $\Delta$ | Train | Test | | | | | | | | | | | | |
| 10 | 2 | G | G | 0.5548 | 0.564 | 0.5608 | 0.5574 | 0.5608 | 0.5597 | 0.5605 | 0.5568 | 0.5571 | **0.576** | 0.5188 | 0.502 |
| 10 | 4 | G | G | 0.4362 | 0.4371 | 0.4277 | 0.4248 | 0.4265 | 0.4331 | 0.4342 | **0.44** | 0.4394 | 0.4345 | 0.3657 | 0.3525 |
| 50 | 4 | G | G | 0.6943 | 0.9286 | 0.9371 | 0.9420 | 0.9394 | 0.9403 | 0.9406 | 0.9394 | **0.9440** | 0.7965 | 0.8128 | 0.758 |
| 50 | 8 | G | G | 0.6403 | 0.8471 | 0.8686 | 0.8617 | 0.8649 | 0.8680 | 0.8651 | **0.8714** | 0.8686 | 0.7048 | 0.7194 | 0.6802 |
| 100 | 8 | G | G | 0.4503 | 0.9623 | 0.9703 | 0.9769 | 0.9769 | **0.9806** | 0.9797 | 0.9797 | 0.9783 | 0.7617 | 0.7931 | 0.7568 |
| 500 | 8 | G | G | 0.2371 | 0.9991 | 0.9991 | 0.9983 | 0.9980 | 0.9989 | 0.9997 | 0.9994 | **1.0000** | 0.7437 | 0.8102 | 0.772 |
| 500 | 8 | GS | G | 0.3980 | 0.7154 | 0.7829 | 0.5966 | 0.7009 | 0.7806 | 0.7777 | 0.8477 | **0.8917** | 0.4157 | 0.4122 | 0.3448 |
| 500 | 8 | G | GM1 | 0.1583 | 0.7760 | 0.8054 | 0.7783 | 0.7883 | 0.7463 | 0.7963 | **0.8374** | 0.8374 | 0.29 | 0.2691 | 0.2317 |
| 500 | 8 | G | GM2 | 0.2237 | 0.9929 | 0.9960 | 0.9954 | 0.9960 | 0.9943 | 0.9954 | 0.9971 | **0.9983** | 0.6531 | 0.6974 | 0.6551 |
| 500 | 8 | MG | MGN | 0.2071 | 0.9657 | 0.9586 | 0.9697 | 0.9551 | 0.9771 | 0.9789 | 0.9740 | **0.9803** | 0.6394 | 0.6471 | 0.6 |

**Table 10:** Test accuracy results on the dynamic tree structure prediction experiments.

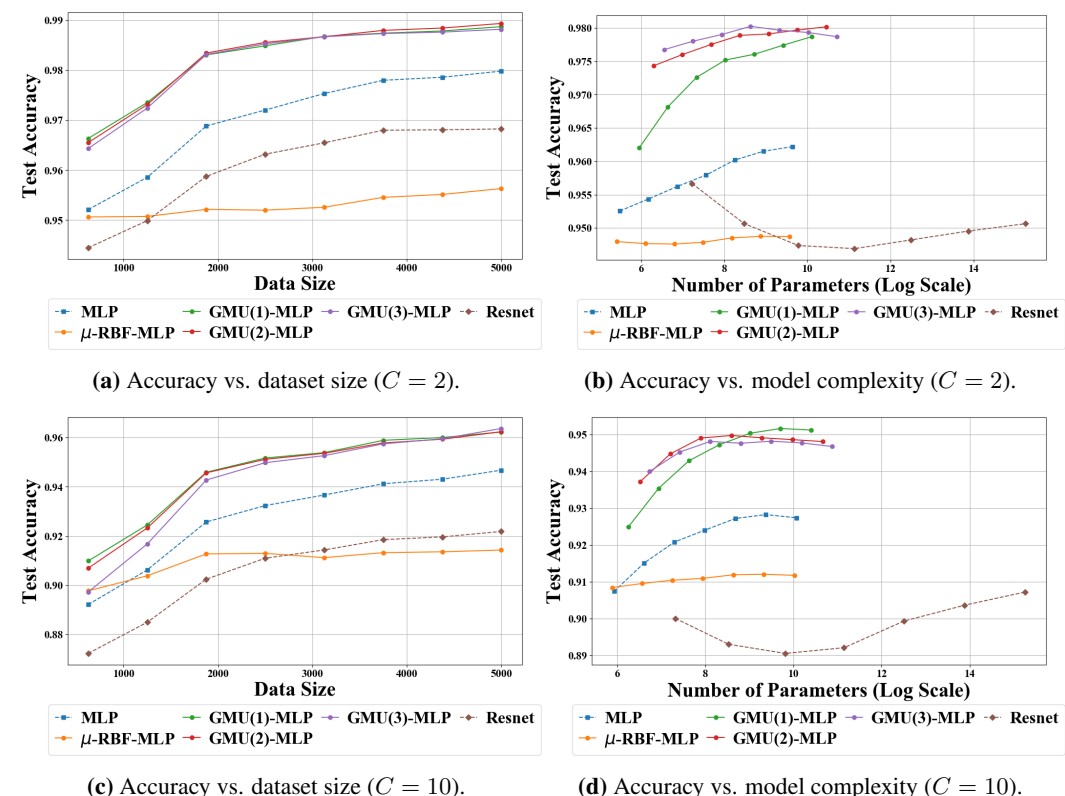

**(a)** Accuracy vs. dataset size ($C = 2$).

**(b)** Accuracy vs. model complexity ($C = 2$).

**(c)** Accuracy vs. dataset size ($C = 10$).

**(d)** Accuracy vs. model complexity ($C = 10$).

**Figure 6:** Scalability studies in two experimental settings in the polynomial boundary sampling experiments. The underlying data distribution is separable via a non-linear decision boundary that is a polynomial function of the data dimensions. We report accuracy trends in response to training dataset size changes in (a) and (c), and in response to parameter count changes in (b) and (d). Please note that only the MLP and Resnet baselines are represented via dashed lines.

| Setup $p$ | GMU(0) | GMU(1) | GMU(2) | GMU(3) | GMU(0)-MLP | GMU(1)-MLP | GMU(2)-MLP | GMU(3)-MLP | Linear | MLP |
|---|---|---|---|---|---|---|---|---|---|---|
| 1 | 0.8223 | 0.8406 | 0.8583 | 0.8643 | **0.9843** | 0.9763 | 0.9743 | 0.97 | 0.983 | 0.9823 |
| 2 | 0.9363 | 0.9363 | 0.9363 | 0.9363 | 0.9816 | **0.9873** | 0.986 | 0.982 | 0.9763 | 0.9801 |
| 3 | 0.7916 | 0.803 | 0.816 | 0.8336 | 0.9596 | **0.975** | 0.974 | 0.9746 | 0.9456 | 0.9606 |

**Table 11:** Test accuracy results on the polynomial boundary sampling experiments.

us separate robustness to noise shape (e.g., symmetric vs skewed) from robustness to multiplicative coupling changes.

**Summary of empirical behavior.** As reported in Table 10, higher-order GMUs consistently outperform lower-order variants and standard feedforward baselines, with gains growing at larger $d$ and under several shift types. Notably, under structure-preserving but distribution-shifting perturbations (e.g., Train: G, Test: GM1/GM2), higher-order GMUs retain substantially more accuracy than MLPs and ResNets. Intuitively, the GMU's internal projection onto low-dimensional generative subspaces is better aligned with the hierarchical linear relations imposed by the tree, producing discriminative signals that are more stable when edge-wise couplings or variances change while the overall SCM topology remains unchanged.

## C.5  POLYNOMIAL BOUNDARY SAMPLING

**Problem Setup:** We consider the scenario where ground truth labels $y$ are assigned based on a set of polynomial functions, with one function corresponding to each class. Given an input $X$, the label is chosen as the one whose polynomial function attains the highest value.

**Definition 6. (*Polynomial Boundary Sampling*)** *Let $X \in \mathbb{R}^d \sim Unif(0,1)^d$, and $y \in \{1, 2, 3, \ldots, C\}$. We define a set of weight matrices $\{W_1, W_2, \ldots, W_d\}$, where $W_i \in \mathbb{R}^{N_y \times p}$. Each class $y_i$ is associated with a polynomial function:*

| Setup | | | GMU(0)-MLP | GMU(1)-MLP | GMU(2)-MLP | GMU(3)-MLP | Linear | MLP | R-[1,1] | R-[2,2] |
|---|---|---|---|---|---|---|---|---|---|---|
| $N_y$ | $d$ | $\sigma_\mu$ | | | | | | | | |
| 2 | 10 | 0.01 | **0.9942** | 0.9888 | 0.9914 | 0.9825 | 0.6474 | 0.9848 | 0.9817 | 0.9782 |
| 10 | 10 | 0.01 | **0.2714** | 0.2608 | 0.244 | 0.2405 | 0.1262 | 0.1908 | 0.1582 | 0.1462 |
| 10 | 10 | 0.1 | **0.8888** | 0.8511 | 0.8254 | 0.8091 | 0.3862 | 0.7685 | 0.7585 | 0.7511 |
| 2 | 100 | 0.01 | **0.7151** | 0.704 | 0.6902 | 0.6851 | 0.5577 | 0.6077 | 0.5694 | 0.5594 |
| 2 | 100 | 0.1 | 0.9414 | 0.9428 | **0.9462** | 0.946 | 0.782 | 0.9248 | 0.9031 | 0.89 |
| 10 | 100 | 0.1 | **0.5145** | 0.4985 | 0.4897 | 0.4897 | 0.2234 | 0.4068 | 0.3214 | 0.2674 |
| 2 | 500 | 0.1 | 0.7377 | 0.7411 | 0.7388 | **0.7454** | 0.6751 | 0.7285 | 0.6905 | 0.6848 |
| 2 | 500 | 1 | 0.986 | 0.9911 | **0.99514** | 0.9951 | 0.9891 | 0.9928 | 0.972 | 0.9794 |

**Table 12:** Test accuracy results on the conditional Gaussian sampling experiments.

$$G(y_i|X) = \sum_{j=1}^{d} W_{ji}[X_j, X_j^2, \ldots, X_j^p]^T. \tag{5}$$

*The label assignment follows a deterministic selection rule as follows:*

$$y^* = \underset{i \in \{1,2,3,\ldots,C\}}{\arg\max} G(y_i|X). \tag{6}$$

*Here, $y^*$ denotes the assigned label based on the evaluated polynomials.*

**Remark 6.** *We consider the binary case, where two polynomial functions define the class boundaries. The label assignment follows: $y = 1$ if $G(y_1|X) - G(y_2|X) > 0$, and $y = 2$ otherwise. Since both $G(y_1|X)$ and $G(y_2|X)$ are polynomials of order $p$, their difference is also a polynomial of order $p$. This establishes a decision boundary that is a polynomial function of order $p$, thus yielding non-linear decision boundaries.*

*This is why we refer to the method as polynomial boundary sampling, as the underlying data distribution is separated by non-linear, polynomial decision boundaries.*

**Experiments**: Each element in the weight matrices $\{W_i\}$ is randomly initialized from $\mathcal{N}(0,1)$. We vary the polynomial order $p$ while keeping $d = 10$ fixed. The following models are compared: GMU($k$), GMU($k$)-MLP-512 (GMU layer with 512 hidden units followed by a linear layer), a single linear layer, and MLP-512. Results are presented in Table 11. Interestingly, we find that as the polynomial order increases, GMU-MLPs of non-zero order show significant improvements on other baselines. To perform a deeper analysis, we conduct scalability studies for the higher order case $p = 3$ as follows.

**Scalability Analysis**: Similar to the sparse neural structure setting, we conduct experiments here to test the performance of networks in response to changes in scale of the dataset and the network complexity itself. We consider two settings. In both settings, we set the order of the polynomial $p = 3$ and input dimensionality $d = 10$. For the first setting, we set $C = 2$ and we set $C = 10$ for the second setting. Due to polynomial decision boundaries $C = 10$ should yield a harder classification problem. The results are shown in Figure 6.

We find that GMU-MLPs show significantly higher generalization performance in this case. For the datasize scaling experiments, we see performance of all networks improve in response to increasing data size, however, the GMU-MLP variants always show a clear improvement. Similarly, for the parameter count versus test accuracy plots, we see that for the same parameter count, GMU-MLPs show significantly higher test accuracy and seem to establish the pareto front. These results highlight the flexibility of GMUs in learning non-linear decision boundaries.

C.6    CONDITIONAL GAUSSIAN SAMPLING

**Problem Setup:** We conduct a simple experiment where the conditional distributions $P(X|y)$ are Gaussian, where $X \in \mathbb{R}^d$ and $y \in \{1, 2, .., C\}$. Specifically, we generate $P(X|y) \sim \mathcal{N}(\mu_y, \Sigma)$. For each dataset, we choose the class-wise mean values by randomly generating them as $\mu_y \sim \mathcal{N}(0, \sigma_\mu^2 I_d)$. Similarly, we pick a randomly generated covariance matrix via $\Sigma = A^T A$ where $A \sim Unif(0,1)^{d \times d}$.

**Experiments:** We pick a range of parameter choices for the sampling process, and compare the following networks: GMU($k$)-MLP-512 (GMU layer with 512 hidden units followed by a linear layer), MLP-512, Resnet-512-[2,1] and Resnet-512-[2,2]. The results are shown in Table 12, for a

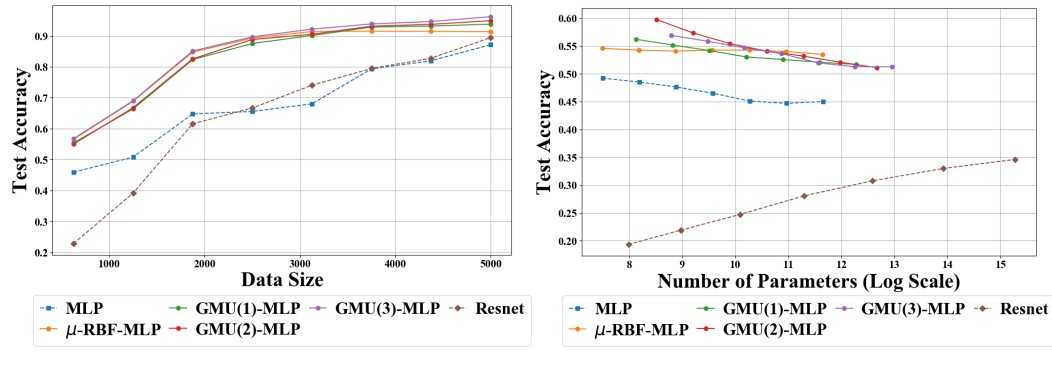

**(a)** Accuracy vs. dataset size.  **(b)** Accuracy vs. model complexity.

**Figure 7:** Scalability studies for the conditional Gaussian sampling experiments. The underlying data distribution of each class is a Gaussian distribution with randomly set covariance matrix and mean. We report accuracy trends in response to training dataset size changes in (a) and in response to parameter count changes in (b). Please note that only the MLP and Resnet baselines are represented via dashed lines.

wide range of parametric choices. We find that overall, $\mu$-RBF networks (GMU(0)-MLP) showcase superior performance. This is unsurprising, as the distance based $\mu$-RBF units are closely aligned with the underlying problem. We also see that for higher dimensionality scenarios, GMU-MLPs can often outperform $\mu$-RBFs. Furthermore, we see that in most cases, GMU-MLP performance for non-zero order is competitive with $\mu$-RBFs, which can also be explained via the universal approximation ability of GMU-MLPs (Proposition 3).

**Scalability Analysis:** Similar to the previous settings, we conduct experiments here to test the performance of networks in response to changes in scale of the dataset and the network complexity itself. We consider the setting where the input dimensionality $d = 100$, $\sigma_\mu = 0.5$ and the number of output classes $C = 10$. The results are shown in Figure 7.

We find that GMU-MLPs and $\mu$-RBFs show better performance than the other baselines. For the datasize scaling experiments, we see performance of all networks improve in response to increasing sample size, and the performance seems to converge for larger sample size. Interestingly, $\mu$-RBF's test accuracy seems to plateau for larger data sizes, whereas GMU-MLPs continue to increase. For the parameter count versus test accuracy plots, we again see that for the same parameter count, $\mu$-RBFs and GMU-MLPs showcase significantly better performance overall. Interestingly, we see that the best performance is actually achieved for lower parameter counts in this setting, and GMU-MLPs showcase slightly better test accuracy than $\mu$-RBFs.

# D  DISCUSSIONS: CONTEXTUALIZING GMUs IN LITERATURE

In this section, we discuss GMUs in the context of existing architectures for supervised learning beyond perceptron-based neural networks. Capsule networks have been proposed in recent years as an alternative learner incorporating feedback. Each hidden node is a *capsule* that contains a vector of activations, and routing iterations adjust the coupling coefficients, directing inputs to hidden units with greater *agreement*. While this introduces a feedback mechanism, it does not function as the internal generative model like GMUs. Furthermore, after routing iterations, the forward computation remains linear, followed by squashing normalization, meaning the complexity bias is still present in the forward pass (Sabour et al., 2017).

Another interesting alternative to multi-layer perceptrons (MLPs) is Kolmogorov-Arnold Networks (KANs, Liu et al. (2024)), which leverage the Kolmogorov-Arnold approximation theorem (Kolmogorov, 1961) to construct more expressive units. However, KANs primarily focus on function approximation and retain a fundamentally feedforward computational paradigm, contrasting with GMUs, which primarily look for dynamic generative structures in the input.

GMUs are also distinct from architectures that rely on weight adaptation mechanisms, such as hypernetworks (Ha et al., 2016) or dynamic convolution (Chen et al., 2020). Unlike these approaches, which dynamically adjust weights, GMUs are designed to reconstruct the input based on projections onto internal $k$-dimensional subspaces, effectively representing reconstruction errors. This makes

GMUs complementary to these models rather than functionally equivalent, as hypernetworks, steerable filters, and similar methods can be used in conjunction with GMUs, just as they would with perceptron or radial basis function (RBF) networks.

Furthermore, GMUs differ structurally from linear self-attention mechanisms (Wang et al., 2020a). Attention mechanisms are fundamentally designed for sequences: they operate by relationally grouping ordered inputs, and even in vision transformers (ViTs, Dosovitskiy (2020)), images are reshaped into patch-based sequences with positional encodings before attention is applied. The goal of attention is thus to dynamically weight interactions across sequence elements. By contrast, GMUs emphasize the structured generative properties of the input and act as direct analogues of classical computational units such as perceptrons or RBFs. Each GMU has its own set of weights and biases and produces an activation response, which in the linear case is simply obtained by computing a $k$-subspace distance followed by an activation function. This generalizes Euclidean distance (used in RBFs) and positions GMUs as a new type of unit at the same architectural level as perceptrons, designed not for sequence modeling but for robust supervised function fitting with explicit theoretical guarantees.

Importantly, just as attention computes similarity via query–key dot products:

$$\text{score}(q, p) = \frac{q^\top p}{\sqrt{d}}, \tag{7}$$

recent variants such as Euclidean Attention and Elliptical Attention (Nielsen et al. (2024)) demonstrate that the score can be defined by alternative distance metrics. In this broader context, one can analogously define a GMU-based attention score using $k$-subspace distance:

$$\text{score}_{k,W}(q, p) = \phi\left(\frac{1}{d} \min_{\theta \in \mathbb{R}^k} \left\| q - p - W^\top \theta \right\|^2\right). \tag{8}$$

As we have shown in this paper, for linear GMUs the $k$-subspace distance admits a closed-form solution (Eq. (3)) and can be efficiently parallelized using tensorized computation with Einstein summations, enabling real-time inference. GMUs can thus be integrated into attention frameworks in two complementary ways: (i) by replacing the similarity score with a generative subspace projection, or (ii) by substituting selected MLP layers within attention blocks with GMU layers. We consider these integration pathways as important directions for future work.

Test-time adaptation (TTA) techniques (Wang et al., 2021) provide another line of work relevant to improving robustness, as they adapt model statistics to distribution shifts encountered during inference. However, these methods do not introduce fundamentally new computational units; rather, they modify existing architectures to better handle domain variability. As we've seen in Appendix B, TTA approaches are complementary to GMUs: while GMUs define a novel unit based on generative subspace projections, TTA can be applied on top of GMU-based networks to further enhance adaptability across domains.

**Generative Capsule Models (GCMs):** Generative Capsule Models (GCMs, Nazabal et al. (2023)) provide an interesting point of comparison to GMUs, since both involve latent linear models for capturing appearance variability. However in GCMs, part-based appearance templates and geometric transformation components are optimized jointly to minimize reconstruction error across samples, yielding structured generative representations. GMUs, by contrast, are not trained to minimize a reconstruction loss at all; their weights are updated exclusively through backpropagation from a supervised objective such as cross-entropy, making them inherently discriminative units. The internal generative model within a GMU serves only as a mechanism for producing its activation signal, and the inferred latent variables ($\theta$) are discarded after the forward pass. This distinction highlights a fundamental difference in purpose: GCMs are designed for unsupervised generative modeling, where the goal is faithful reconstruction of inputs, while GMUs are designed for supervised function fitting, where the goal is predictive performance. Moreover, GCMs rely on more computationally expensive variational inference schemes (such as ELBO optimization) to handle both appearance and geometric variability, whereas GMUs work with a one-shot solution that emerges from a MAP approximation under a CL-SCM sampling framework, allowing them to integrate efficiently into standard supervised pipelines. Thus, while both models have latent linear structures that do sample-wise inference, their optimization paradigms and roles in learning are fundamentally different.

### D.1 CAN KOLMOGOROV ARNOLD NETWORKS (KANS) EFFICIENTLY RE-PARAMETERIZE GMUS?

In this section we examine whether Kolmogorov–Arnold Networks (KANsLiu et al. (2024)) can serve as efficient re-parameterizations of GMUs. Although KANs and GMUs are motivated by entirely different principles; KANs by universal function approximation, GMUs by Bayes optimality to switching SCM based data generation; we ask whether KANs can nevertheless parameterize GMUs efficiently. In theory, KANs can represent *any* continuous function; here we outline the construction that would allow such equivalence and then test whether the resulting KAN design is feasible in practice. This sets the stage for evaluating if the universality promised by the Kolmogorov–Arnold representation can also reproduce the function of GMU architectures.

#### D.1.1 KOLMOGOROV–ARNOLD REPRESENTATION AND CANONICAL KANS

The Kolmogorov–Arnold theorem (KAT) states that any continuous function $f : [0,1]^d \to \mathbb{R}$ can be represented as

$$f(x_1, \ldots, x_d) = \sum_{q=1}^{2d+1} \Phi_q \left( \sum_{j=1}^{d} \Psi_{qj}(x_j) \right), \tag{9}$$

where each inner function $\Psi_{qj}$ is univariate in its coordinate $x_j$ and each outer function $\Phi_q$ is univariate. Canonical KAN layers operationalize this template by replacing linear edge weights with learnable univariate functions (often spline parameterizations) and aggregating additively before a single outer nonlinearity:

$$h_i(x) = g_i \left( \sum_{j=1}^{d} f_{ij}(x_j) + b_i \right), \tag{10}$$

with $f_{ij} : \mathbb{R} \to \mathbb{R}$ univariate (e.g., spline bases) and $g_i$ a simple scalar activation (identity or smooth). This induces an axis-aligned inductive bias and per-coordinate interpretability.

#### D.1.2 GMU STRUCTURE AND THE SUM-OF-SQUARES EXPONENTIAL FORM

The linear GMU unit computes an activation from a projector-induced residual norm:

$$\mathrm{gmu}(X) = \phi \left( \frac{1}{d} \left\| (I - P_W) \left( \frac{X - b}{\eta} \right) \right\|^2 \right), \qquad P_W = W^\top (W^\top)^\dagger, \tag{11}$$

where $\phi(t) = \exp(-t)$ in our default choice. Writing $Z = (X - b)/\eta$ and letting $\{v_r\}_{r=1}^d$ be the row vectors of $I - P_W$.

$$\| (I - P_W) Z \|^2 = \sum_{r=1}^{d} (v_r^\top Z)^2, \tag{12}$$

so that

$$\mathrm{gmu}(X) = \exp \left( -\frac{1}{d} \sum_{r=1}^{d} (v_r^\top Z)^2 \right). \tag{13}$$

This form is an exponential of a *sum of squared linear forms*, where the mixing directions $\{v_r\}$ are geometrically tied to the learned subspace via $P_W$.

#### D.1.3 WHY A SINGLE CANONICAL KAN LAYER CANNOT REPRODUCE GMUS

A canonical KAN layer enforces a separable preactivation $\sum_j f_j(X_j)$ with one outer nonlinearity $g(\cdot)$. While setting $g(t) = t^2$ can realize *one* squared linear form $(\sum_j u_j X_j)^2$ by choosing $f_j(X_j) = u_j X_j$, a single layer cannot, under the univariate-per-edge constraint, produce a *sum of multiple squared linear forms* $\sum_{r=1}^{d} \left( \sum_j v_{rj} Z_j \right)^2$ and then apply a single exponential to that aggregate. The inner sum in canonical KANs is scalar; producing multiple squared linear forms requires multiple

distinct preactivations prior to aggregation, which exceeds the expressivity of one KAN layer with strictly univariate edges.

### D.1.4 Two layer KAN equivalence in theory but not in practice

For $m$ GMU outputs with input dimension $d$ (zero bias, unit scale), each GMU $i$ computes

$$y_i(X) = \exp\Big(-\tfrac{1}{d}\sum_{r=1}^{d}(v_r^{(i)\top}Z^{(i)})^2\Big), \qquad Z^{(i)} = \tfrac{X - b_i}{\eta_i}.$$

A minimal two-layer KAN replication instantiates, for every output $i$ and residual direction $r$,

$$\text{Layer 1:} \quad s_r^{(i)}(X) = \sum_{j=1}^{d} v_{rj}^{(i)} Z_j^{(i)}, \qquad h_r^{(i)}(X) = \big(s_r^{(i)}(X)\big)^2,$$

and then performs grouped aggregation in Layer 2. In the worst case $k_i = 0$, the total number of Layer-1 units is $N_{\mathrm{L1}} = m \cdot d$, and the grouping is explicit:

$$\text{Layer 2:} \quad S_i(X) = \sum_{r=(i-1)d+1}^{id} h_r(X), \qquad y_i(X) = \exp\Big(-\tfrac{1}{d}S_i(X)\Big), \quad i = 1, \ldots, m.$$

This construction is exact in theory, but extremely impractical in practice:

- **Strict Activation functions.** The construction requires Layer 1 to apply a squaring nonlinearity and Layer 2 to apply a single exponential. These choices are fixed and leave no room for the spline-based activation learning that defines canonical KANs.

- **Strict Connectivity.** Layer 2 must partition the $m \cdot d$ Layer-1 units into $m$ disjoint groups of size $d$ and sum only the units in each group to form the corresponding GMU output. This strict wiring is non-native to canonical KANs.

- **Unit and parameter blowup.** Example: $m = d = 100 \Rightarrow N_{\mathrm{L1}} = 10{,}000$ Layer-1 units. Each unit carries a length-$d$ mixing vector, yielding $\approx m\,d^2 = 1{,}000{,}000$ parameters.

- **Non-native constraints.** Exact equivalence requires the per-output mixing vectors $\{v_r^{(i)}\}$ to originate from a Cholesky-decomposable matrix so that the squared linear forms correspond to a valid residual norm. Canonical B-spline KANs do not enforce such structured parameterizations, making equivalence fragile under standard training.

**In Summary:** While a two-layer KAN can replicate the GMU parameterization on paper (as it can replicate *any* continuous function by design), it demands highly specific connectivity, non-standard activation choices, and massive overparameterization. The only way to avoid overparameterization is via structured external constraints such as Cholesky factorization which are not native to KANs, and even in that case the other issues still remain. Canonical KANs therefore cannot reproduce GMUs in practice without heavy specification and computational inefficiency.

## D.2 Comparing with Capsule Networks

Capsule networks and GMUs are motivated by fundamentally different principles. Capsules emphasize *self-organization via feedback and routing*: their dynamic routing mechanism iteratively adjusts coupling coefficients between capsules to capture part–whole relationships in data (Hinton et al., 2018; Sabour et al., 2017). GMUs, by contrast, have no such feedback component. They are designed as generative computational units that measure similarity through $k$-subspace distances, providing robust supervised function fitting with explicit theoretical guarantees.

Importantly, capsule routing typically computes similarity between a lower-level capsule $u_i$ and a higher-level capsule $v_j$ via a dot product or cosine similarity:

$$c_{ij} \propto u_i^\top v_j, \tag{14}$$

where $c_{ij}$ denotes the coupling coefficient. Recent work has explored distance-based routing strategies, such as K-Means routing for text classification (Zhang et al., 2019) and for complex

image analysis (Wang et al., 2020b). These approaches replace dot-product similarity with Euclidean-type distances in the clustering step, showing that capsule routing can be generalized beyond inner products:

$$c_{ij}^{\text{Euc}} = \phi\big(-\|u_i - v_j\|^2\big). \tag{15}$$

In this broader context, one can analogously define a GMU-based capsule similarity using the $k$-subspace distance:

$$\text{sim}_{k,W}(u_i, v_j) = \phi\left(\frac{1}{d} \min_{\theta \in \mathbb{R}^k} \|u_i - v_j - W^\top \theta\|^2\right), \tag{16}$$

where $W \in \mathbb{R}^{k \times d}$ parameterizes a learnable generative subspace, $b$ is a bias term, and $\phi(\cdot)$ is an activation function. This generalizes the Euclidean case ($k = 0$) and positions GMUs as a principled alternative similarity measure for capsule routing.

As we have shown in this paper, for linear GMUs the $k$-subspace distance admits a closed-form solution (Eq. 3) and can be efficiently parallelized using tensorized computation with Einstein summations, enabling real-time inference. GMUs can actually be integrated into capsule frameworks in two complementary ways: (i) by replacing the similarity score with a generative subspace projection, or (ii) by substituting the reconstruction MLP layers within capsule networks with GMU layers.

Thus fundamentally, capsule networks answer a different question than GMUs: they are about *feedback and routing*, while GMUs are about *generative similarity and robust activation*. Importantly, these principles are not mutually exclusive. As we show here, capsule routing can occur under GMU-based similarity computations, which represents an important direction of future work with GMUs.

# E  DETAILS ON GMU FUNCTION AND INTEGRATION

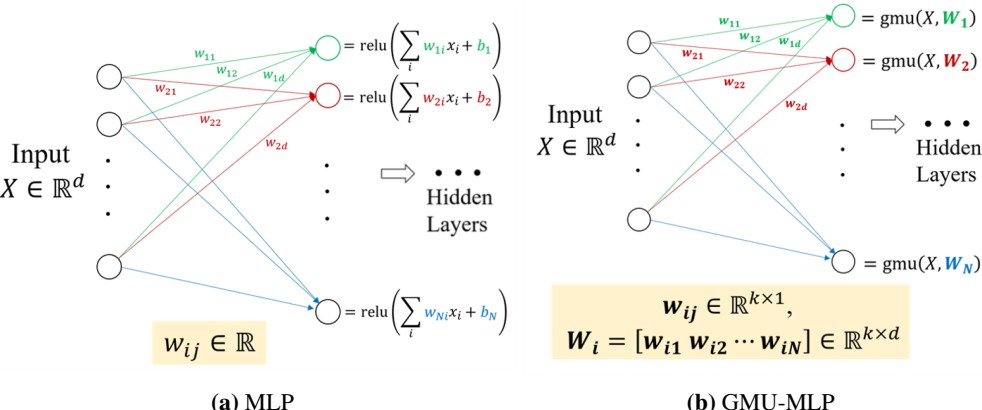

**(a)** MLP          **(b)** GMU-MLP

**Figure 8:** MLPs versus GMU-MLPs: Demonstration of how GMUs are integrated into architectures. Here, we replace the first network layer with GMU computational units.

## E.1  INTEGRATING GMUS INTO ARCHITECTURES

**GMU-MLPs** As shown in Figure 8, GMU integration into an MLP is direct, and only involves replacing the computational units with GMU units. Like perceptron units, linear GMU units have their own weights and biases, which are visualized in Figure 8. The network is trained via standard backpropagation, which in our case is handled by Pytorch's autograd framework.

**GMU-CNNs** When replacing a convolution layer with GMUs, we replace each linear convolutional operation with a convolutional GMU. The operation of a convolutional GMU is shown in Figure 9. $X_{ij}^a$ represents the input patch of size $a \times a$, which is the input to the GMU, the output of which is simply the GMU's output $\text{gmu}(X_{ij}^a, W)$, where $W$ are its internal weights. As the operation is a convolution, this function is repeated across all patches centered at all $(i, j)$ pairs in the input image, yielding an output image of the same size (assuming padding is used). Mathematically, we can represent the convolutional GMU operation as $\text{gmu}(., W) * X$.

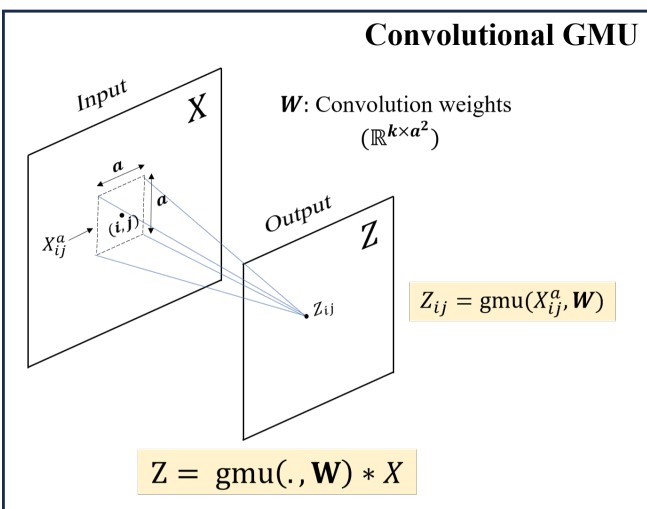

**Figure 9:** Diagram showing the operations within a convolutional GMU. Here, the convolutional weights $W$ are applied to the input $X$ yielding the output $Z$. For the purposes of this figure, we assume that the convolution is appropriately padded such that $X$ and $Z$ are of the same size.

### E.2   LINEAR GMUs: CLOSED FORM EXPRESSION

Here, we summarize the least square based approach that yields the closed-form expression of a linear GMU's output in equation 3. The minimization in equation 1 can be represented as

$$\min_{\theta \in \mathbb{R}^k} \left\| \frac{X - b}{\eta} - W^T \theta \right\| \tag{17}$$

Note that as $X \in \mathbb{R}^d$ (consider as a column vector), the above minimization represents the least squares solution to a set of $d$ equations, one for each dimension of $X$. The solution to it is as follows

$$\arg\min_{\theta \in \mathbb{R}^k} \left\| \frac{X - b}{\eta} - W^T \theta \right\| = (WW^T)^{-1} W \left( \frac{X - b}{\eta_X} \right) \tag{18}$$

Note that here $\frac{X-b}{\eta_X}$ denotes a matrix of size $d \times 1$. With this, each GMU unit's output in equation 1 can be represented as

$$\phi \left( \frac{1}{\sigma d} \left\| \frac{X - b}{\eta} - W^T (WW^T)^{-1} W \left( \frac{X - b}{\eta} \right) \right\|^2 \right) \tag{19}$$

Note that the above is the output of a single GMU, which only depends on the input to the unit $X$, the weights and biases $W, b$, and other parameters such as the normalizer $\eta$, the scaling parameter $\sigma$ and the dimensionality of input $d$.

## F   ADDITIONAL EMPIRICAL DETAILS

### F.1   SYNTHETIC EXPERIMENTS

Please refer to Figure 2(b) for the detailed jargon in defining GMU types.

**Sparse Linear Structure Prediction:** For the GMU($k$) variants, we used a unit without normalization and bias. $\phi(z) = -\log z$ (to counter-act the softmax function that follows) and $W = var(\mathbb{R}^{k \times d})$. For the out-of-distribution (ood) columns, we set the training $\theta_{l_y(i)}$ ranges to either between $Unif(0, 0.5)$ or $Unif(0.5, 1)$ chosen at random. For the test data, we change the range for each $\theta_{l_y(i)}$ in such a manner that if its training configuration was $Unif(0, 0.5)$ it is set to $Unif(0.5, 1)$ and vice-versa. This ensures that at test-time the network sees values of the latent generating variables which it hasn't seen before.

**Dynamic Tree Structure Prediction:** To generate the random trees, first, we sample the root node from a uniform distribution over all dimensions, and then sample left and right nodes uniformly from the rest of the dimensions. The process is then repeated for the children of all leaf nodes. Once the children of all leaf nodes at a certain depth have been sampled, the process is repeated for the children of the subsequent depth of the tree.

For the GMU($k$)-MLP variants, for the GMU units, we used units with normalization $\eta = \sigma(X)$ and bias. $\phi(z) = \sqrt{1-z}$ and $W = [var(\mathbb{R}^{k-1 \times d}); J_{1,d}]$. To have a fair comparison, each datapoint was also normalized using zero-mean and unit variance for the MLP variants.

**Sparse Neural Structure Sampling**: For the GMU units in the GMU($k$)-MLP variants, we used units without normalization, but non-zero bias. $\phi(z) = -\log z$ and $W = var(\mathbb{R}^{k \times d})$.

**Polynomial Boundary Sampling:** For both the GMU($k$)-MLP and the GMU($k$) variants, for the GMU units, we used units without normalization, but non-zero bias. $\phi(z) = e^{-z}$ and $W = var(\mathbb{R}^{k \times d})$.

**Conditional Gaussian Sampling:** For the GMU units in the GMU($k$)-MLP variants, we used units without normalization, but non-zero bias. $\phi(z) = e^{-z}$ and $W = var(\mathbb{R}^{k \times d})$.

### F.2 TABULAR EXPERIMENTS

In addition to the GMU results shown in the main paper (GMU-S), we test another low-parameter configuration of GMUs called GMU-D. Using the jargon in Figure 2(b), the details are as follows.

**GMU-S:** We choose non-zero bias $b$ and the normalization factor $\eta = \sigma_{X-b}$. The activation function is set to $\phi(z) = e^{-z}$. The weights are $W = var(\mathbb{R}^{k \times d})$

**GMU-D:** We choose non-zero bias $b$ and the normalization factor $\eta = \sigma_{X-b}$. The activation function is set to $\phi(z) = e^{-z}$. The weights are $W = [w, w^2, ..w^k], w : var(\mathbb{R}^d)$.

| Dataset | #Ins./#Feat. | XGB. | ASK-G. | TabN. | AutoGL. S | Resnet-530 (1.2M) | GMU-S (1.195M) | GMU-D (1.159M) |
|---|---|---|---|---|---|---|---|---|
| credit-g | 1000 / 21 | 0.6893 | 0.7119 | 0.6119 | 0.6964 | 0.6972 | 0.7036 | **0.7262** |
| anneal | 898/39 | 0.8542 | **0.9000** | 0.8425 | 0.8000 | 0.8546 | 0.8610 | 0.8583 |
| kr-vs-kp | 3196/37 | **0.9985** | **0.9985** | 0.9325 | 0.9969 | 0.9969 | 0.9969 | 0.9969 |
| mfeat-factors | 2000/217 | 0.9800 | 0.9750 | 0.9725 | 0.9800 | **0.9850** | 0.9800 | 0.9825 |
| vehicle | 846/19 | 0.7497 | 0.8017 | 0.7965 | 0.8379 | 0.8446 | **0.8793** | 0.8493 |
| kc1 | 2109/22 | 0.6685 | 0.6335 | 0.5252 | 0.6727 | **0.7104** | 0.6866 | 0.6824 |
| phoneme | 5404/6 | 0.8797 | 0.8834 | 0.8682 | 0.8394 | 0.8959 | 0.8882 | **0.9039** |
| cnae-9 | 1080/857 | **0.9491** | 0.9352 | 0.8935 | 0.9259 | 0.9398 | 0.9398 | 0.9306 |
| blood | 748/5 | 0.6228 | 0.6499 | 0.6433 | **0.6725** | 0.6304 | **0.6718** | 0.5987 |
| Australian | 690/15 | **0.8972** | 0.8859 | 0.8528 | 0.8825 | 0.8479 | 0.8726 | 0.8692 |
| car | 1728/7 | 0.9238 | **1.0000** | 0.9870 | 0.9968 | **1.0000** | **1.0000** | **1.0000** |
| segment | 2310/20 | **0.9372** | 0.9307 | 0.9178 | 0.9199 | 0.9254 | 0.9307 | 0.9351 |
| jasmine | 2984/145 | **0.8055** | 0.7888 | 0.7669 | 0.8005 | 0.7438 | 0.7520 | 0.7688 |
| sylvine | 5124/21 | 0.9307 | 0.9336 | 0.8360 | **0.9375** | 0.9250 | 0.9268 | 0.9268 |
| adult | 48842/15 | 0.7982 | 0.7983 | 0.7716 | **0.8056** | 0.7652 | 0.7735 | 0.7622 |
| nomao | 34465/119 | 0.9687 | **0.9722** | 0.9543 | 0.9642 | 0.9582 | 0.9599 | 0.9575 |
| bank | 45211/17 | 0.7266 | 0.7228 | 0.7064 | **0.7948** | 0.7261 | 0.7382 | 0.729 |
| jungle_chess | 44819/7 | 0.8733 | 0.8307 | 0.7343 | 0.9302 | 0.9686 | 0.9773 | **0.9897** |
| volkert | 58310/181 | 0.6417 | 0.6343 | 0.5941 | 0.7020 | 0.6766 | 0.7003 | **0.7096** |
| helena | 65196/28 | 0.2199 | 0.2114 | 0.1903 | **0.2712** | 0.2213 | 0.2206 | 0.2137 |
| connect-4 | 67557/43 | 0.7237 | 0.7265 | 0.7205 | **0.7562** | 0.7369 | 0.7535 | 0.743 |
| higgs | 98050/29 | 0.7294 | 0.7293 | 0.7204 | **0.7380** | 0.6684 | 0.6781 | 0.6808 |
| numerai28.6 | 96320/22 | 0.5236 | **0.5242** | 0.5160 | 0.5171 | 0.5066 | 0.5146 | 0.5071 |
| walking-activity | 149332/5 | 0.6162 | 0.6276 | 0.5680 | 0.6080 | 0.6203 | **0.6322** | 0.6026 |
| ldpa | 164860/8 | **0.9901** | 0.6895 | 0.5482 | 0.5302 | 0.6991 | 0.6777 | 0.6621 |
| aloi | 108000/129 | 0.9534 | 0.1353 | 0.9359 | **0.9742** | 0.9617 | 0.9684 | 0.9622 |
| skin-seg | 245057/4 | 0.9997 | 0.9997 | 0.9996 | 0.9997 | 0.9997 | 0.9996 | 0.9996 |
| **Wins/losses/ties** | **GMU-S vs** | 13/11/3 | 13/11/3 | 23/3/1 | 11/13/3 | 18/4/5 | – | |
| **Wilcoxon** $p$ | **GMU-S vs** | 0.3533 | 0.4750 | 2.3e-05 | 0.732 | 0.0089 | 0.224 | |
| **Wins/losses/ties** | **GMU-D vs** | 12/14/1 | 12/13/2 | 22/4/1 | 12/13/2 | 13/8/5 | – | |
| **Wilcoxon** $p$ | **GMU-D vs** | 1.000 | 0.968 | 3.2e-04 | 0.657 | 0.407 | 0.224 | |

**Table 13:** Performance Comparison of Various Models on Different Datasets. Parameter counts are given in brackets in the top-row. Please note that we scaled the original Resnet-512 to Resnet-530 to match parameter counts with the GMU-Resnet models.

The networks were trained in the same manner as in Kadra et al. (2021), using weighted cross-entropy loss, and for evaluation we also report the balanced accuracy, same as them. We compare GMU-Resnet-512-[1,1] with Resnet-512-[1,1]. We set the same hyperparameters for all experiments, and don't perform any additional hyperparameter optimization. In addition to the GMU configuration described in Section 4.2, we also construct another GMU with less trainable parameters as follows.

Note that the other approaches' results are after extensive hyperparameter optimization using BOHB (Falkner et al., 2018). Note that Kadra et al. (2021) uses a different Shaped Resnet architecture and therefore we don't directly compare with their MLP results, and we find in some datasets our Resnet performs significantly better than theirs and vice-versa. Furthermore the MLP+C approach in Kadra et al. (2021) employs an extensive suite of regularization approaches, including data augmentation, so we don't include their results for this study. Similarly, the MLP-Dropout in Kadra et al. (2021) also uses hyperparameter optimization for the dropout levels and locations for each dataset.

We add a single dropout layer (of 0.2) at the penultimate layer for both Resnet-512-[1,1] variants, as we found it led to more stable training overall. . Apart from this, there is no regularization or data augmentation performed, and networks are trained in the same manner for all datasets. The average parameter count for each approach is given below the network names in Table 13.

The categorical variables within the data were one-hot encoded, and the other variables were normalized to the range (0,1), with the statistics computed only from the training split. The training-test splits are exactly the same as in Kadra et al. (2021), which is an 80-20 split, with the same random seed values set by them. This was made possible by their code, and the fact that each dataset corresponded to a specific task as numbered in Table 9 in Kadra et al. (2021).

### F.3 Vision Experiments

**GMU Details:** Using the jargon in Figure 2(b), the details are as follows. We set bias to zero $b = 0$ and the normalization factor $\eta = \sigma_X$. The activation function is set to $\phi(z) = e^{-z}$. The weights are $W = [var(\mathbb{R}^{k-1 \times d}); J_{1,d}]$, same as what was used in the dynamic tree structure prediction experiment.

**Architectures:** We train a four-layer CNN for MNIST and Fashion-MNIST, with the architecture 64C(3,Padding=1)-2MP-128C(3,Padding=1)-2MP-128C(3,Padding=1)-2MP(Padding=1)-128C(3,Padding=1)-4MP-FC128-FC10, where C(k,Padding=j) denotes $k \times k$ convolutional layers with a padding of $j$ in either direction, MP denotes max pooling layers, and FC denotes fully connected layers. The first convolutional layer is replaced with convolutional GMUs of order 3. For CIFAR-10, we use VGG-16 as the base network, replacing the first convolutional layer with convolutional GMU-3 units while keeping the same number of output nodes. Similarly, for smallNORB, we resize images to 64x64 and train a four-layer CNN: 64C(5,Padding=0)-2MP-128C(5,Padding=0)-2MP-128C(3,Padding=0)-2MP-128C(3,Padding=0)-4MP-FC10, with the first convolutional layer replaced with convolutional GMUs. Note that for smallNORB, we test GMUs of varying orders (1 and 3), the results of which are reported in Table 4 of the main paper.

For SVHN, we train a ResNet-18 architecture, replacing the first convolutional layer with convolutional GMU units of order 3 and kernel size 7x7, maintaining the original receptive field. We applied standard data augmentation (for all baselines) involving rotations and translations. Note that no data augmentation was applied to any of the other datasets.

### F.4 Optimization

We summarize the optimization details for the experiments on the Tabular and Vision datasets. We use Adam optimizer for all experiments. All networks are trained for 200-300 epochs, with an exponential learning rate scheduler $lr = init_{lr} \times 0.5^{epoch/50}$, where $init_{lr}$ is the initial learning rate. For the GMU runs, we found that picking the network with the lowest training loss yielded more stable performance, so we stick with this approach for all experiments in the Tabular and Vision datasets. For the MLP/Resnet runs, we didn't see any such improvement, so we use the standard approach of training for a fixed number of epochs. A reason why this would be the case is that GMU-MLPs'/GMU-CNNs' losses are slightly more fluctuating in nature than MLP/Resnet's losses, so if the training stopped at an epoch which had higher fluctuations of training loss, that could be a detriment to performance.

### F.5 Computation Times

We document the per-sample inference times of GMU-based architectures and compare to the standard counterparts in each case. First, we report the per-sample inference times (in milliseconds) in all our tabular datasets tested in Table 14. Overall, although the standard counterparts are faster, we find that GMUs only take about 0.0017 milliseconds (ms) on average for per-sample inference, retaining their feasibility for real-time inference and fast training.

| Dataset | Dimension | GMU-Resnet Time (ms) | Resnet Time (ms) |
|---|---|---|---|
| credit-g | 63 | 0.0181 | 0.0036 |
| anneal | 72 | 0.0171 | 0.0041 |
| kr-vs-kp | 74 | 0.0188 | 0.0045 |
| mfeat-factors | 216 | 0.0149 | 0.0036 |
| vehicle | 18 | 0.0183 | 0.0044 |
| kc1 | 21 | 0.0214 | 0.0051 |
| phoneme | 5 | 0.0167 | 0.0040 |
| cnae-9 | 856 | 0.0398 | 0.0067 |
| blood transfusion | 4 | 0.0205 | 0.0048 |
| australian | 42 | 0.0221 | 0.0054 |
| car | 21 | 0.0177 | 0.0042 |
| segment | 16 | 0.0197 | 0.0047 |
| jasmine | 280 | 0.0159 | 0.0037 |
| sylvine | 20 | 0.0182 | 0.0043 |
| adult | 105 | 0.0154 | 0.0036 |
| nomao | 174 | 0.0156 | 0.0037 |
| bank marketing | 51 | 0.0156 | 0.0037 |
| jungle chess | 6 | 0.0153 | 0.0036 |
| volkert | 180 | 0.0155 | 0.0037 |
| helena | 27 | 0.0155 | 0.0037 |
| connect-4 | 126 | 0.0155 | 0.0036 |
| higgs | 28 | 0.0154 | 0.0037 |
| numerai28.6 | 21 | 0.0154 | 0.0037 |
| walking activity | 4 | 0.0153 | 0.0037 |
| ldpa | 14 | 0.0153 | 0.0036 |
| aloi | 128 | 0.0185 | 0.0036 |
| skin segmentation | 3 | 0.0153 | 0.0036 |
| **Average** | - | 0.0179 | 0.0041 |

**Table 14:** Per-sample Inference Times in tabular datasets (in milliseconds)

| Dataset | GMU-1 | GMU-3 | GMU-5 | GMU-8 | CNN |
|---|---|---|---|---|---|
| MNIST | - | 0.0183 | - | - | 0.0088 |
| Fashion-MNIST | - | 0.0193 | - | - | 0.0045 |
| CIFAR-10 | - | 0.0232 | 0.0232 | 0.0244 | 0.0089 |
| Small-NORB | 0.0440 | 0.0456 | - | - | 0.0111 |
| ImageNet (ResNet-50) | 14.263 | 13.085 | 13.728 | 14.395 | 8.841 |

**Table 15:** Per-sample inference times (milliseconds) in vision datasets.

We also report the per-sample inference times for the GMU-CNNs tested in MNIST, Fashion-MNIST, CIFAR-10 in Table 15. To see the feasibility of scaling GMUs to Imagenet level datasets, we also include per-sample inference times of a pre-trained Resnet-50 on 224x224x3 sized Imagenet inputs. Following the main paper, the first layer of the original networks were replaced by convolutional GMUs. We find that the compute times stay insignificant for all datasets. For Imagenet, the compute times for GMU-Resnets are larger ($> 10$ ms), but still feasible for real-time inference, and comparable to the standard Resnet.

## F.6 PARAMETER COUNT COMPARISONS

We report the parameter counts of baseline architectures and their GMU-augmented counterparts across all our real dataset experiments. GMUs are only substituted in the first convolutional or dense layer, so the overall increase in parameters remains negligible. For tabular experiments, GMU-S and GMU-D were configured to keep parameter counts close to the ResNet-530 baseline.

As seen in Table 16, the parameter growth from GMU substitution is insignificant relative to the total model size. Even in deeper architectures such as VGG16 and ResNet18, the increase is less than one percent, while in smaller CNNs the absolute growth remains on the order of only a few thousand parameters. Note that for tabular experiments, the baseline Resnet was scaled to 530 hidden units for fair comparison, after which the parameter counts are similar, with GMU-D having slightly lower counts than the baseline.

| Dataset | Model | Baseline Params | GMU Variant | GMU Params | % Change |
|---|---|---|---|---|---|
| MNIST / Fashion-MNIST | Shallow CNN | 0.39M | GMU-3 | 0.39M | +0.3% |
| CIFAR-10 | VGG16 | 14.7M | GMU-3 / GMU-8 | 14.7M / 14.7M | +0.02% / +0.08% |
| SVHN | ResNet18 | 11.2M | GMU-3 | 11.2M | +0.17% |
| smallNORB | Shallow CNN | 0.52M | GMU-1 / GMU-3 | 0.52M / 0.52M | 0% / +0.6% |
| Tabular (Average) | ResNet-530 | 1.20M | GMU-S / GMU-D | 1.19M / 1.16M | -0.8% / −3.4% |

Table 16: Baseline vs GMU parameter counts across datasets and models. GMUs are applied only in the first layer

## G  Proofs of Theoretical Results

**Proposition 1.** *Let the random variables $(X, Y)$ follow the ECI sampling process, and let $S = \{(X_1, Y_1), \ldots, (X_n, Y_n)\}$ be i.i.d. samples from this distribution. Consider a single-layer neural network with computational units $\mathrm{pu}_i(X, \mathbf{W}, \mathbf{b}) := \sum_{j=1}^{d} W_{ij} x_j + b_i, \quad i \in \{1, \ldots, M\}$, followed by a softmax layer producing output $P_{\mathrm{soft}}(Y|X, \mathbf{W}, \mathbf{b})$. Let $\mathcal{L}_{\mathrm{CE}}$ denote the empirical cross-entropy loss and assume it is uniformly bounded over the support of $P(Y|X)$. Then, for every $(\mathbf{W}^*, \mathbf{b}^*) \in \arg\min_{\mathbf{W}, \mathbf{b}} \mathcal{L}_{\mathrm{CE}}(S)$, $\lim_{n \to \infty} \mathrm{KL}\left(P(Y|X) \, \| \, P_{\mathrm{soft}}(Y|X, \mathbf{W}^*, \mathbf{b}^*)\right) = 0$ almost surely.*

*Proof.* Using Bayes' rule,

$$P(Y|X) = \frac{P(Y) \prod_{j=1}^{d} P(x_j|Y)}{\sum_{i=1}^{M} P(Y=i) \prod_{j=1}^{d} P(x_j|Y=i)},$$

and substituting the ECI form $P(x_j|Y=i) = \frac{1}{Z_{ij}} \exp(\alpha_{ij} x_j)$ gives

$$P(Y|X) = \frac{P(Y) \prod_{j=1}^{d} \frac{1}{Z_{ij}} \exp(\alpha_{ij} x_j)}{\sum_{i=1}^{M} P(Y=i) \prod_{j=1}^{d} \frac{1}{Z_{ij}} \exp(\alpha_{ij} x_j)}.$$

This is of the same functional form as the softmax output, since the neural network logits are defined by

$$\mathrm{pu}_i(X, \mathbf{W}, \mathbf{b}) = \sum_{j=1}^{d} W_{ij} x_j + b_i,$$

and by setting $W_{ij} = \alpha_{ij}$ and $b_i = \log P(Y) - \sum_{j=1}^{d} \log Z_{ij}$ we have $\log P(Y|X) \propto \mathrm{pu}_i(X, \mathbf{W}, \mathbf{b})$, so that

$$P_{\mathrm{soft}}(Y|X, \mathbf{W}^*, \mathbf{b}^*, S) = \frac{\exp(\mathrm{pu}_i(X))}{\sum_{i=1}^{M} \exp(\mathrm{pu}_i(X))}.$$

Since the cross-entropy loss $\mathcal{L} = -\mathbb{E}_{P(Y|X)}[\log P_{\mathrm{soft}}(Y \mid X)]$ differs from the KL divergence only by the entropy of $P(Y \mid X)$, minimizing cross-entropy is equivalent to minimizing $D_{\mathrm{KL}}(P(Y \mid X) \, \| \, P_{\mathrm{soft}}(Y \mid X))$. Because both $P(Y \mid X)$ and $P_{\mathrm{soft}}(Y \mid X)$ share the same functional form under the ECI assumption, the minimum is attained when this KL divergence is zero. By the strong law of large numbers, since the samples are i.i.d. and the loss is uniformly bounded, the empirical cross-entropy converges almost surely to its population expectation as $n \to \infty$. Thus, $\lim_{n \to \infty} D_{\mathrm{KL}}(P(Y \mid X) \, \| \, P_{\mathrm{soft}}(Y \mid X, \mathbf{W}^*, \mathbf{b}^*)) = 0$ almost surely.

$\square$

**Proposition 2.** *Let the random variables $(X, Y)$ follow the CL-SCM sampling process with equal class priors, and let $S = \{(X_1, Y_1), \ldots, (X_n, Y_n)\}$ be i.i.d. samples from this distribution. Consider a single-layer neural network with computational units $\mathrm{g}_i(X, \mathbf{W}, \sigma) := -\frac{1}{\sigma^2} \min_\theta \mathbb{E}[\|f(\theta, W_i) - X\|^2]$ for $i \in \{1, \ldots, M\}$, followed by a softmax layer producing output $P_{\mathrm{soft}}(Y \mid X, \mathbf{W}, \sigma)$. Let $\mathcal{L}_{\mathrm{CE}}$ denote the empirical cross-entropy loss and assume it is uniformly bounded over the support of $P(Y \mid X)$. Then, for every $(\mathbf{W}^*, \sigma^*) \in \arg\min_{\mathbf{W}, \sigma} \mathcal{L}_{\mathrm{CE}}(S)$, we have $\lim_{n \to \infty} \mathrm{KL}(P(Y \mid X) \, \| \, P_{\mathrm{soft}}(Y \mid X, \mathbf{W}^*, \sigma^*)) = 0$ almost surely.*

**Remark 1.** *Please note that we assume the classes are equiprobable a priori, which is not there in the submitted main paper version of this result.*

*Proof.* Under the CL–SCM sampling process, the generative model is

$$X = f(\theta, W_i) + \epsilon, \quad \epsilon \sim \mathcal{N}(0, \sigma^2 I), \quad \theta \sim \mathcal{U}([-1, 1]^k).$$

As we're assuming a MAP estimate of posteriors, we have

$$P(X \mid Y = i) = \max_\theta P(X \mid Y = i, \theta)P(\theta).$$

Since $P(\theta)$ is uniform (i.e. constant), this reduces to maximizing $P(X \mid Y = i, \theta)$, and we have

$$P(X \mid Y = i) = \frac{1}{V_\theta(\sigma\sqrt{2\pi})^d} \exp\Big(-\frac{1}{2\sigma^2} \min_\theta \|X - f(\theta, W_i)\|^2\Big),$$

where $V_\theta$ is the volume of $\theta$'s support space. By Bayes' rule we can write $P(Y = i \mid X) = P(Y = i)P(X \mid Y = i)/\sum_{j=1}^M P(Y = j)P(X \mid Y = j)$.

Given $g_i(X, \mathbf{W}, \sigma)$ we note that $\exp g_i(X, \mathbf{W}, \sigma)$ has the same function form as the numerator in the expression of $P(Y = i|X)$ (after cancelling out the constants), and subsequently, a single-layer neural network with softmax computes

$$P_{\text{soft}}(Y = i \mid X, \mathbf{W}^*, \sigma^*, S) = \frac{\exp\big(g_i(X, \mathbf{W}, \sigma)\big)}{\sum_{j=1}^M \exp\big(g_j(X, \mathbf{W}, \sigma)\big)},$$

which is the same function form as $P(Y = i|X)$. Since minimizing the cross-entropy loss is equivalent to minimizing $D_{\text{KL}}(P(Y \mid X) \| P_{\text{soft}}(Y \mid X))$, and as $P(Y \mid X)$ and $P_{\text{soft}}(Y \mid X)$ share the same functional form under the CL-SCM assumption, the minimum is attained when this KL divergence is zero. By the strong law of large numbers, since the samples are i.i.d. and the loss is uniformly bounded, the empirical cross-entropy converges almost surely to its population expectation as $n \to \infty$. Thus, $\lim_{n\to\infty} D_{\text{KL}}(P(Y \mid X) \| P_{\text{soft}}(Y \mid X, \mathbf{W}^*, \sigma^*)) = 0$ almost surely.

$\square$

**Proposition 3.** *(from Park and Sandberg (1991)) We are given a GMU-MLP, with GMU units in equation 1 specified as: $k = 0, \eta = 1$, $f(\theta, W)$ is such that $\exists W s.t. f(\theta, W) = 0$ and $\phi(z)$ is any integrable bounded function such that $\int \phi(x)dx \neq 0$. Then, this GMU-MLP can approximate any function $f \in L^p(\mathbb{R}^d)$.*

*Proof.* First, we note that the set of functions approximable by the GMU-MLP under the constraint $\exists W s.t. f(\theta, W) = C$ is a subset of the set of functions that the GMU-MLP can approximate without that constraint. Next, we note that when we set the weights of the GMU such that $f(\theta, W) = C$, setting $\sigma'^2 = \sigma^2 d$, and $\eta = 1$ yields an RBF unit as the bias $b$ can be adjusted as $b' = b - C$ to yield the original RBF form. By Theorem 1 in Park and Sandberg (1991), it is known that RBF-based networks achieve universal approximation in $L^p(\mathbb{R}^d)$, thus proving the result. $\square$

**Theorem 2.** *Let $E(X_1, X_2) = \|X_1 - X_2\|$, we can show that $\gamma_E(d + 1) < \gamma_E(d)$.*

*Proof.* Since $X_1, X_2 \sim \mathcal{N}(0, I_d)$, we have $X_1 - X_2 \sim \mathcal{N}(0, 2I_d)$, so that $\|X_1 - X_2\| = \sqrt{2}\chi_d$, where $\chi_d$ is a random variable with the chi distribution with $d$ degrees of freedom. Its mean is $\mathbb{E}[\chi_d] = \sqrt{2}\frac{\Gamma\left(\frac{d+1}{2}\right)}{\Gamma\left(\frac{d}{2}\right)}$ and its variance is $\text{Var}(\chi_d) = d - \left(\sqrt{2}\frac{\Gamma\left(\frac{d+1}{2}\right)}{\Gamma\left(\frac{d}{2}\right)}\right)^2$. Therefore, the mean of $E(X_1, X_2) = \|X_1 - X_2\|$ is $\mu_E = \sqrt{2}\,\mathbb{E}[\chi_d] = 2\frac{\Gamma\left(\frac{d+1}{2}\right)}{\Gamma\left(\frac{d}{2}\right)}$ and its standard deviation is $\sigma_E = \sqrt{2\big(d - 2\frac{\Gamma\left(\frac{d+1}{2}\right)^2}{\Gamma\left(\frac{d}{2}\right)^2}\big)}$.

Defining the CV as $\gamma_E(d) = \sigma_E/\mu_E$, we obtain

$$\gamma_E(d)^2 = \frac{\frac{d}{2}\,\Gamma\left(\frac{d}{2}\right)^2}{\Gamma\left(\frac{d+1}{2}\right)^2} - 1.$$

Thus, showing that $\gamma_E(d) < \gamma_E(d-1)$ is equivalent to proving

$$\frac{\Gamma\left(\frac{d+1}{2}\right)\Gamma\left(\frac{d-1}{2}\right)}{\Gamma\left(\frac{d}{2}\right)^2} > \sqrt{\frac{d}{d-1}}, \tag{20}$$

The above follows from the result using Gurland's ratio in equation 1.1 of Tian and Yang (2021), where it states:

$$\frac{\Gamma(x)\Gamma(x+2u)}{\Gamma(x+u)^2} > 1 + \frac{u^2}{x} \tag{21}$$

Setting $x = (d-1)/2$ and $u = 1/2$, we get:

$$\frac{\Gamma\left(\frac{d+1}{2}\right)\Gamma\left(\frac{d-1}{2}\right)}{\Gamma\left(\frac{d}{2}\right)^2} > 1 + \frac{1}{2(d-1)} \tag{22}$$

Squaring the RHS, we get: $\left(1 + \frac{1}{2(d-1)}\right)^2 = 1 + \frac{1}{d-1} + \frac{1}{4(d-1)^2} > \frac{d}{d-1}$. Thus we have,

$$\frac{\Gamma\left(\frac{d+1}{2}\right)\Gamma\left(\frac{d-1}{2}\right)}{\Gamma\left(\frac{d}{2}\right)^2} > \sqrt{\frac{d}{d-1}}, \tag{23}$$

yielding the result.

$\square$

**Proposition 4.** *We are given the $k$-subspace distance $S_{k,W}(X_1, X_2)$ from $X_1$ to $X_2$. With this, first, we note that $\gamma_{S_{0,W}}(d) = \gamma_E(d)$, where $E$ denotes the Euclidean distance. Then, we have that $\gamma_{S_{k+1,W}}(d) > \gamma_{S_{k,W}}(d)$, and thus $\gamma_{S_{k,W}}(d) > \gamma_E(d)$.*

*Proof.* The proof directly follows by realizing that for a fixed $k$-subspace, the closest distance to a point is equivalent to the squared root of sum of square of $d - k$ dimensions $x_1, x_2, .., x_{d-k}$ in the Euclidean space, where each dimension $x_i \sim \mathcal{N}(0, 1)$ as the original data is also distributed this way.

This holds simply because one can rotate the space to align its unit vectors with the orthogonal directions of the $k$-subspace, leaving only the other $d - k$ to have degrees of freedom.

With this, it directly follows that $\gamma_{S_{k,W}}(d) = \gamma_E(d-k) < \gamma_E(d-k-1) = \gamma_{S_{k+1,W}}(d)$. And it naturally follows that $\gamma_{S_{k,W}}(d) = \gamma_E(d-k) > \gamma_E(d)$. $\square$

**Proposition 5.** *We consider the case where the RV $X$ has low intrinsic dimensionality. Let $X \in \mathbb{R}^d$ be generated as: $X = \sum_{i=1}^{k_{ID}} a_i \boldsymbol{v_i}$, $a_i \sim \mathcal{N}(0, 1)$, where $\{\boldsymbol{v_i}\}_{i=1}^{k_{ID}}$ are orthonormal vectors in $\mathbb{R}^d$, and $k_{ID} < d$ is the intrinsic dimensionality of $X$. Let us set $W = [\boldsymbol{v_1}, .., \boldsymbol{v_k}]$ to be the first $k$ orthonormal vectors. Then, we have that $\gamma_E(d) = \gamma_E(k_{ID})$ and $\gamma_{S_{k,W}}(d) = \gamma_E(k_{ID} - k)$.*

*Proof.* It is trivial to show that $\gamma_E(d) = \gamma_E(k_{ID})$. The result directly follows from the fact that $X$ lies in a $k_{ID}$ dimensional linear subspace with all Gaussian distributed dimension components $\mathcal{N}(0, 1)$. Thus, we have that $\sigma[S(X_1, X_2)] = \sigma_{X_1', X_2' \sim \mathcal{N}(0, I_d)}[S(X_1', X_2')]$, where $X_1'$ and $X_2'$ are the $k_{ID}$ dimensional. Ssimilarly, it follows that $\mu[S(X_1, X_2)] = \mu_{X_1', X_2' \sim \mathcal{N}(0, I_d)}[S(X_1', X_2')]$. Thus, we obtain $\gamma_E(d) = \gamma_E(k_{ID})$.

Let $X_1 = \sum_{i=1}^{k_{ID}} a_i \boldsymbol{v_i}$ and $X_2 = \sum_{i=1}^{k_{ID}} a_i' \boldsymbol{v_i}$. As $W$ is constructed using the first $k$ components of the linear subspace that contains $X$, the k-subspace distance $S_{k,W}(X_1, X_2)$ can be expressed as:

$$S_{k,W}(X_1, X_2) = \min_{\beta} ||\sum_{i=1}^{k_{ID}} a_i \boldsymbol{v_i} + \sum_{i=1}^{k} \beta_i \boldsymbol{v_i} - \sum_{i=1}^{k_{ID}} a_i' \boldsymbol{v_i}|| \tag{24}$$

$$= \min_{\beta} \sqrt{\sum_{i=1}^{k} (a_i + \beta_i - a_i')^2 + \sum_{i=k+1}^{k_{ID}} (a_i - a_i')^2} \tag{25}$$

$$= \sqrt{\sum_{i=k+1}^{k_{ID}} (a_i - a_i')^2} \tag{26}$$

where the minimum is reached when $a_i + \beta_i - a_i' = 0$ for all $i = \{1, 2, .., k\}$. Subsequently we note that the resulting distance is simply the distance between two $k_{ID} - k$ dimensional vectors, the entries of which are generated via the Gaussian distribution $\mathcal{N}(0, 1)$. Thus, the $k$-subspace distance here is equivalent to taking a distance in a lower dimensional space of dimension $k_{ID} - k$, the CV of which is $\gamma_E(k_{ID} - k)$. This proves the second part of the result. $\qquad\square$

