# OpenReview forum: "Learning Structured Dependencies using Generative Computational Units"
_ICLR.cc/2026/Conference — ICLR 2026 Conference Desk Rejected Submission_

### Official Review · Reviewer_cDvd · 2025-10-23

**Soundness:** 2
**Presentation:** 3
**Contribution:** 2
**Rating:** 4
**Confidence:** 2

**Summary:**

This paper introduces the Generative Matching Unit (GMU), a new neural computational unit designed to capture structured dependencies in data through an internal generative model. Key contributions include: (1) formalizing GMU variants (e.g., linear GMUs) with closed-form expressions and theoretical guarantees of universal approximation and Bayesian optimality under structured sampling processes; (2) demonstrating GMU's robustness to the curse of dimensionality compared to RBF units via k-subspace distance metrics; and (3) empirical validation showing improved generalization and out-of-distribution robustness in tabular (e.g., GMU-ResNets) and vision tasks (e.g., GMU-CNNs).

**Strengths:**

1. The paper is novel.
2. The paper is very well-written and clearly structured.
3. The paper is completed with theory, analysis, and experiments.

**Weaknesses:**

1. The paper's most significant limitation is the decision to place GMUs only in the first layer of the networks tested. While justified by computational concerns, this severely restricts the assessment of GMUs as a general-purpose computational unit.
2. While testing on 27 tabular datasets is thorough, the vision benchmarks are relatively standard and small-scale. The paper does not demonstrate that GMUs scale effectively to larger, more complex datasets, such as ImageNet.
3. The paper would benefit from a more systematic ablation of the GMU's hyperparameters, particularly the order $k$ and the activation function $\phi$.

**Questions:**

See the Weakness section.

---

> ### Author Response · Authors · 2025-12-04
> **Response**
>
> We thank the reviewer for their comments and suggestions. Below we address their specific questions and concerns.
>
> > The paper's most significant limitation is the decision to place GMUs only in the first layer of the networks tested. While justified by computational concerns, this severely restricts the assessment of GMUs as a general-purpose computational unit.
>
>
> We place GMUs only in the first hidden layer because they are a **natural evolution of radial basis units**, which in RBF networks are almost always restricted to a single hidden layer (Broomhead & Lowe, 1988; Park & Sandberg, 1991; Bishop, 1995). **Stacking RBF layers is not standard practice**, as it produces nested exponentials (e.g., $e^{\sum_i w_i \, e^{\|W_i - X_i\|}}$) that risk vanishing/exploding gradients (Goodfellow et al., 2016). This is also highlighted in Wurzberger & Schwenker (2024), which reiterates that **universal approximation is already guaranteed with one RBF layer**. In fact, their work is among the first to explore stacking RBFs in multiple layers, which ends up requiring **specialized schemes** and represents a separate research direction. In the same way, our choice to restrict GMUs to the first hidden layer is principled and consistent, and our extensive synthetic and real experiments confirm its effectiveness. **Stacking GMU layers will also require further care** like in Wurzberger & Schwenker (2024), and would represent a separate research direction.
>
> > While testing on 27 tabular datasets is thorough, the vision benchmarks are relatively standard and small-scale. The paper does not demonstrate that GMUs scale effectively to larger, more complex datasets, such as ImageNet.
>
> We recognize the reviewer’s concern regarding ImageNet. Our vision experiments were not designed to compete on large‑scale benchmarks, but to **rigorously test whether GMUs can be integrated into convolutional architectures and whether they improve robustness under distribution shifts**. In this respect, GMU‑CNNs **consistently outperform standard CNNs (including VGG and Resnet configurations) across MNIST, Fashion‑MNIST, CIFAR‑10, and smallNORB**, and show statistically significant gains even when combined with state‑of‑the‑art test‑time adaptation methods (**Appendix B**). This is achieved with essentially the **same amount of parameters (Appendix F.6)**. In total, we evaluate **38 corruption configurations**, providing a comprehensive robustness analysis.
>
> Beyond vision, our evaluation spans **28 tabular datasets** (13 with ≥50,000 samples and 6 with >100,000 samples; see Table 13) and **30 synthetic classification datasets** probing different aspects of generalization, each with in‑ and out‑distribution variants (**≈60 result sets**). Together, these experiments demonstrate that GMUs scale effectively across diverse modalities and dataset sizes, and that their benefits are not confined to small‑scale settings. Please note that most of the **synthetic studies are provided in Appendix C**.
>
> As **Wurzberger & Schwenker (2024)** note in their work on deep RBF networks, scaling locally‑tuned units to very deep or ImageNet‑scale architectures is a distinct research challenge. The central contributions of our work are theoretical and foundational: we establish conditions under which **computational units achieve Bayes optimality**, yielding GMUs as a principled extension of RBF units, derive **closed‑form expressions for linear GMUs**, prove their **universal approximation** properties, and analyze their **robustness to the curse of dimensionality.** These results establish GMUs as a principled extension of RBF units. The accompanying synthetic and real‑data experiments validate these theoretical insights across diverse modalities. Extending GMUs to ImageNet‑scale benchmarks is an exciting future direction, but the present work establishes the theoretical and empirical groundwork necessary for that step.
>
> > The paper would benefit from a more systematic ablation of the GMU's hyperparameters, particularly the order  and the activation function .
>
>
> Please note that in our synthetic experiments and several real‑data experiments, we **already report results with different GMU configurations**, including variations in order and degree, while keeping the activation function choice fixed within each experiment to ensure methodological consistency and to avoid dataset‑specific tuning and bias. Nonetheless, **additional ablations**, particularly for all of the vision datasets, will appear in the final version of the paper to further strengthen the empirical analysis.

---

### Official Review · Reviewer_ZBLM · 2025-10-28

**Soundness:** 2
**Presentation:** 2
**Contribution:** 2
**Rating:** 4
**Confidence:** 3

**Summary:**

This paper introduces the Generative Matching Unit (GMU), a new type of neural computational unit that embeds generative model
within each neuron. Instead of performing a linear mapping Wx+b as in a perceptron, a GMU infers latent parameters through an
internal function f(θ,W) and computes its activation from the minimal reconstruction error between the generated and observed
inputs. The authors argue that this design allows GMUs to capture structured dependencies in data and achieve “Bayes-optimal”
behavior under class-wise structural causal models (CL-SCMs). The authors primarily study linear GMUs, a simplified version that
shows analytical similarity to RBF units while claiming richer representational capacity. They further integrate these GMUs into
standard neural architectures, constructing MLP- and CNN-based networks in which conventional layers are replaced by GMU
layers to evaluate their effectiveness on both tabular and vision benchmarks.

**Strengths:**

1. The idea of reinterpreting neural computation units from a Bayesian–generative perspective is conceptually interesting.
2. The definition of the GMU is clearly structured and easy to follow. The mathematical formulation and analysis are
generally sound and clearly presented.

**Weaknesses:**

1. The proposed GMU is functionally similar to RBF units —it computes a distance between the input and an internal template, followed by a nonlinear mapping. The internal “latent generative model” serves only as a reparameterized projection, not a true generative process.
2. The proposed GMU introduces a significantly higher computational cost compared to standard RBF units. While this theoretically allows
more expressive representations, the paper does not demonstrate any scenario where such additional complexity leads to substantial benefits.
3. The theoretical discussion of Bayes-optimality, ECI, and SCM essentially reformulates the Naive Bayes assumption without offering
substantial new insight. This section feels overly long and contributes little to the actual development of the GMU.
4. The evaluation is limited. Comparisons are restricted to RBF networks and standard CNN/ResNet baselines, without including more recent probabilistic or energy-based architectures. The experiments are conducted only on small-scale tabular and vision datasets, which limits the generality of the results and makes it difficult to assess the scalability of the proposed approach.
5. Furthermore, the experiments replace only the first layer of standard networks with GMUs, limiting the evaluation to shallow settings and preventing a deeper investigation of whether the proposed generative properties hold in multi-layer architectures.

**Questions:**

1. Could the proposed GMU behavior be approximated by stacking conventional neurons or RBF units to form equivalent subspace
projections? If so, what unique representational benefit does GMU provide beyond reparameterizing existing architectures?

2. The GMU introduces an internal pseudo-inverse computation, which can be expensive. Could the authors provide complexity analysis or
discuss practical settings where the additional cost is justified?

3. All experiments use only the linear version of the GMU. Have the authors explored nonlinear variants, and if so, what challenges were
encountered in optimization or computational cost?

---

> ### Author Response · Authors · 2025-12-04
> **Response-1**
>
> We thank the reviewer for their comments and suggestions. Below we address their specific questions and concerns.
>
>
> > The proposed GMU is functionally similar to RBF units —it computes a distance between the input and an internal template, followed by a nonlinear mapping. The internal “latent generative model” serves only as a reparameterized projection, not a true generative process.
>
>
> The reviewer notes that GMUs appear functionally similar to RBF units, computing a distance followed by a nonlinear mapping, and suggests that the internal latent generative model serves only as a reparameterized projection. We clarify that this is **not merely a heuristic construction: GMUs are derived directly from latent variable generative models.**
>
> In particular, when GMUs use **linear latent variable models (LLVMs)**, the internal computations emerge from **MAP inference** under well-defined generative priors:
> $\theta \sim \mathcal{U}(-1, 1), \quad \epsilon \sim \mathcal{N}(0, \sigma^2 I),$
> with the generative process being:
> $X = W^\top \theta + b + \epsilon.$
>
> As shown in Definition 2 and Proposition 2 of our paper, MAP inference in this generative setup yields the residual form used in GMUs. Thus, in the linear case, GMUs reduce to computing the projection residual of $X$ onto the subspace spanned by $W$, consistent with inference in LLVMs. This demonstrates that the GMU formulation is grounded in a true generative process, not just an ad‑hoc projection.
>
> For nonlinear latent generative models $f(\theta, W)$, the same principle applies, but the residual is computed against a nonlinear manifold rather than a linear subspace. This generalization distinguishes GMUs from RBF units: while RBFs are fixed similarity measures, GMUs retain generative semantics through latent variables and priors, extending naturally to nonlinear settings.
>
>
>
> > The proposed GMU introduces a significantly higher computational cost compared to standard RBF units. While this theoretically allows more expressive representations, the paper does not demonstrate any scenario where such additional complexity leads to substantial benefits.
>
> **GMUs do not introduce significantly high computational cost compared to standard RBF** units. Rather, the computational overhead is actually negligible for two main reasons:
>
> - As shown in eq. (3), we **bypass the minimization** as it is simply equivalent to finding the pseudo-inverse, the computation of which is already optimized in GPUs.
> - GPUs also enable **efficient parallelization** of pseudo-inverse computations using tensorized representations of data with Einstein summations (see **SRModule_Conv function in https://anonymous.4open.science/r/GMU-B86C/Demo/predict_fmnist.py**). This yields real-time inference (please see **Table 14/15 for detailed computation times including Imagenet** and **https://streamable.com/anox59**).
>
> Beyond computational efficiency, the **empirical benefits of GMUs are consistently demonstrated** across synthetic, tabular, and vision experiments. GMU‑based architectures outperform baselines both **in‑distribution and out‑of‑distribution**. This is the central contribution of our work: showing that the generative grounding of GMUs yields tangible improvements without incurring prohibitive computational cost.
>
> > The theoretical discussion of Bayes-optimality, ECI, and SCM essentially reformulates the Naive Bayes assumption without offering substantial new insight. This section feels overly long and contributes little to the actual development of the GMU.
>
> The purpose of Section 3 is **not to re‑state Naive Bayes**, but to make explicit the generative assumptions that motivate GMUs. Naive Bayes is often ambiguously described as either a generative model or simply a classifier under conditional independence. To remove this ambiguity, we introduce the precise notion of **exponential conditionally independent (ECI) sampling**, which formalizes the factorization with exponential conditionals and linear powers. This makes clear what the underlying generative process is.
>
> Importantly, Proposition 1 only serves as a **motivational benchmark** for us: it shows that a perceptron is Bayes‑optimal under ECI sampling. The real contribution of the section is that under more structured generative settings such as **class‑wise latent SCM sampling (CL‑SCM, Definition 2)**,  a **new Bayes-optimal unit emerges (Proposition 2)**. The CL‑SCM is not ECI-based or Naive Bayesian as it does not impose any conditional independence and reflects realistic, structured causal assumptions.
>
> Crucially, the **residual form of GMUs emerges directly from Proposition 2**. Without this derivation, the GMU computation would appear arbitrary. This section therefore provides the theoretical foundation that connects GMUs to Bayes‑optimality under structured generative models, which is substantially different and doesn’t relate to reformulating Naive-Bayes.

---

> ### Author Response · Authors · 2025-12-04
> **Response-2**
>
> > The evaluation is limited. Comparisons are restricted to RBF networks and standard CNN/ResNet baselines, without including more recent probabilistic or energy-based architectures. The experiments are conducted only on small-scale tabular and vision datasets, which limits the generality of the results and makes it difficult to assess the scalability of the proposed approach.
>
> We would like to clarify that GMUs are **neither probabilistic nor energy‑based models**. GMU computations are direct and deterministic, with residual error minimized at the level of internal units rather than via an architectural energy function. Comparisons to probabilistic or energy‑based approaches are therefore not directly relevant.
>
> We would like to clarify: 13 out of our 28 Tabular datasets have **$\gtrsim$ 50,000 samples**, while 6 of them have **more than 100,000 samples**, so we do not consider our Tabular experiments to be limited to small scale data (see Table 13). The objective of vision experiments was primarily to test robustness, and we did extensive test-time corruption analysis for all four datasets we tested, including comparing with benchmark corruption handling approaches such as test-time adaptation (**Appendix B**).
> In total, we test on **28 Tabular datasets** that span a very diverse range/type of data, **30 synthetic classification tasks** that each probe a different aspect of generalization, each having in- and out-distribution variants (i.e.**$\approx$ 60 sets of results**), and in total **38 vision dataset configurations** overall including each dataset and its associated corruption.
>
>
> > Furthermore, the experiments replace only the first layer of standard networks with GMUs, limiting the evaluation to shallow settings and preventing a deeper investigation of whether the proposed generative properties hold in multi-layer architectures.
>
> We place GMUs only in the first hidden layer because they are a **natural evolution of radial basis units**, which in RBF networks are almost always **restricted to a single hidden layer** (Broomhead & Lowe, 1988; Park & Sandberg, 1991; Bishop, 1995). Stacking RBF layers is not standard practice, as it produces nested exponentials (e.g., $ e^{\sum_i w_i \, e^{\|W_i - X_i\|}}$) that risk **vanishing/exploding gradients** (Goodfellow et al., 2016). This is also highlighted in Wurzberger & Schwenker (2024), which reiterates that universal approximation is already guaranteed with one RBF layer. In fact, their work is **among the first to explore stacking RBFs** in multiple layers, which ends up requiring specialized schemes and represents a **separate research direction**. In the same way, our choice to **restrict GMUs to the first hidden layer** is principled and consistent, and our extensive synthetic and real experiments confirm its effectiveness. Stacking GMU layers will also require further care like in Wurzberger & Schwenker (2024), and would represent a separate research direction.

---

> ### Author Response · Authors · 2025-12-04
> **Response-3**
>
> > Could the proposed GMU behavior be approximated by stacking conventional neurons or RBF units to form equivalent subspace projections? If so, what unique representational benefit does GMU provide beyond reparameterizing existing architectures?
>
> No, the **GMU behaviour cannot be approximated by stacking conventional neurons or RBF units** or any combination of them. This is simply because each GMU, if viewed from the perspective of an RBF, uses “centers” which result from a computation of the pseudoinverse of an arbitrary weight matrix $W$ followed by projecting the normalized input $(X-b)/\eta$. Thus these **centers are not fixed, but are dynamic**, being dependent both on the normalized inputs and the pseudoinverse of $W$. Thus, RBF networks cannot replicate them. ReLU activated neural networks by default cannot exactly replicate them because reLU-activated networks can only approximate RBF networks (**Opschoor, J. A. 2020**) and RBFs are a specialized case of GMUs when the weights $W=0$.
>
> What can be used to replicate the behaviour, however, is a **very specific combination of layers with strong parametric and architectural constraints**. To replicate the behaviour of a GMU layer with multiple outputs (e.g., 100 units), the following will be needed:
> 1. **Replicated Normalization layers (100 times)**
>    - Each one requires its own affine normalization $(X-b_i)/\eta_i$.
>    - These are per-output operations with distinct learned biases $b_i$ and choices of $\eta_i$, not shared across outputs.
>    - For a layer with 100 outputs, this means 100 separate normalization branches.
>
> 2. **Linear Projection layers with Cholesky factored weights**
>    - Each branch applies a projection operator $I - W_i^T(W_i^T)^\dagger$ to the normalized input.
>    - This requires parameterizing weights through the pseudo-inverse operation, producing one projection per output.
>    - With 100 outputs, there are 100 parallel projection operators.
>
> 3. **Parallel RBF layers centered at 0**
>    - Each projection residual is fed into a fixed RBF neuron centered at the origin.
>    - Each RBF computes $\exp(-\|A_i z_i\|^2)$ (or a scaled variant), operating in parallel across all outputs.
>    - Thus, for 100 outputs, there are 100 parallel RBFs, each tied to its own projection residual.
>
> This construction can in principle reproduce the desired behaviour, but under **strong architectural and weight constraints** and with significant parameter overhead, as detailed above. Even so, this approach introduces substantial memory overhead, and our standard GMU implementation is significantly more straightforward, parametrically efficient and direct.
>
> #### Opschoor, J. A., Petersen, P. C., & Schwab, C. (2020). Deep ReLU networks and high-order finite element methods. Analysis and Applications, 18(05), 715-770.
>
> > The GMU introduces an internal pseudo-inverse computation, which can be expensive. Could the authors provide complexity analysis or discuss practical settings where the additional cost is justified?
>
> The **additional time cost is not very significant**, as extensively discussed in **Appendix F.5**. Given the significant gains across our experiments: 28 Tabular datasets, 30 synthetic classification settings, and 38 vision dataset configurations, we believe the slightly added computational cost is not of concern.

---

> ### Author Response · Authors · 2025-12-04
> **Response-4**
>
> > All experiments use only the linear version of the GMU. Have the authors explored nonlinear variants, and if so, what challenges were encountered in optimization or computational cost?
>
> Non-linear GMUs can take many forms, but unlike the linear case they lose the closed-form solution (eq. (3)) and become slower to compute. Therefore we don’t study them in this work. However, it is quite possible to extend our work to the case where the generative models are non-linear and avoid significant increase in computation time. This is because there exist approaches which can **one-shot approximate solutions to non-linear fitting problems** (Moré 1978, Rahimi, A., & Recht, B. 2007). We thus consider this one of the significant directions of future work with GMUs.
> Given this however, a natural question is whether linear **GMUs remain useful when the underlying data manifolds are non-linear**. Our experiments in **Appendix C.3 (sparse neural structure sampling) and C.5 (polynomial boundary sampling)** directly address this. In both cases, the data is generated from explicitly non-linear manifolds, yet linear GMUs consistently outperform baselines:
>
> - **C.3: Sparse Neural Structure Sampling**
>   Data generated by 2-layer tanh networks yields class-specific non-linear manifolds. Linear GMUs scale better with dataset size and achieve near Pareto-optimal accuracy vs. complexity, outperforming MLP, RBF, and ResNet baselines.
>
> - **C.5: Polynomial Boundary Sampling**
>   Data is separated by polynomial decision boundaries of order 1–3. Linear GMUs (and GMU-MLPs) consistently outperform linear layers and standard MLPs, with gains increasing as polynomial order rises. GMUs generalize better across binary and multi-class settings, often establishing the Pareto front.
>
> **In short, our tests confirm: even when the data manifolds are non-linear, linear GMUs exploit local linearity and yield better fit and generalization while retaining efficiency. This highlights the capability of linear GMUs, and addresses why non-linear GMUs are a natural but not a necessary extension at this point.**
>
> #### Moré, J. J. (1978). The Levenberg–Marquardt Algorithm: Implementation and Theory. In Numerical Analysis (Lecture Notes in Mathematics, vol. 630).
> #### Rahimi, A., & Recht, B. (2007). Random Features for Large-Scale Kernel Machines. In Advances in Neural Information Processing Systems (NeurIPS).

---

### Official Review · Reviewer_7BWj · 2025-10-29

**Soundness:** 2
**Presentation:** 3
**Contribution:** 1
**Rating:** 2
**Confidence:** 3

**Summary:**

In this work, the author proposed generative matching units (GMU) as a computational unit in deep learning models to enhance the generalizability and capture structural dependency in data. The authors also proved that GMU are universal approximators and are efficient and robust. GMUs are also designed to avoid the curse of dimensionality. On tabular datasets and vision datasets, GMU also demonstrate higher classification accuracy than baselines.

**Strengths:**

1. This work is clearly presented. The logic is linear and the core mechanism of proposed GMUs is well articulated.

2. Theories developed and discussed in this work are compact. How GMU can resolve the curse of dimensionality issue is sufficiently scrutinized in theory. Extensive experiments demonstrate the effectiveness of proposed GMU to avoid the curse of dimensionality issue. They indicate the comprehensiveness and coherence of this work. Theoretically and empirically, the discussion of the curse of dimensionality issue is of high quality.

**Weaknesses:**

1. Although attention units are mentioned in Introduction, it is not clearly and comprehensively discussed in related works and methodologies. As an important and ubiquitous computational unit, the attention unit should be considered as a baseline in terms of generalizability, robustness to OOD cases and efficiency for the proposed GMU.

2. The significance of this work definitely needs to be clearly and explicitly studied and articulated. Nowadays, computational units are not of such high significance in the era of LLMs, if the computational units proposed cannot answer the pressing questions, such as security, reasoning, etc.

3. The technical novelty and originality of this work also requires to be highlighted and stated with clarity. The generative Bayesian computation methods in recent two years are not referred and compared, such as [1].

[1] Maria Nareklishvili, Nicholas Polson, and Vadim Sokolov, Generative Bayesian Computation for Causal Inference, arXiv, vol. 2306.16096, 2024.

**Questions:**

1. Why is it still necessary to propose and study GMU as universal approximators in this era on top of self-attention? Is it possible to answer the question why foundation models can hallucinate and solve this problem? Can GMU explain and improve the interpretability of foundation models?

2. Compared with newly proposed computational units such as self-attention, KAN or capsule networks, does GMU demonstrate higher generalizability and robustness theoretically or empirically, especially against OOD scenarios?

3. For GMU, could the authors give an example of nonlinear GMU? Why in this paper nonlinear GMU is not chosen as a computational unit in deep learning models for study?

---

> ### Author Response · Authors · 2025-12-04
> **Response-1**
>
> We thank the reviewer for their comments and suggestions. Below we address their specific questions and concerns.
>
> > Although attention units are mentioned in Introduction, it is not clearly and comprehensively discussed in related works and methodologies. As an important and ubiquitous computational unit, the attention unit should be considered as a baseline in terms of generalizability, robustness to OOD cases and efficiency for the proposed GMU.
>
> Response:
> We have more comprehensive discussions in **Appendix D, D.1 and D.2**, which has been significantly expanded upon to more clearly elaborate the structural, functional, and purpose‑driven motivations of different computational units. In particular, attention mechanisms are fundamentally **designed for sequences**: they operate by relationally grouping ordered inputs, and even in vision transformers (ViTs), images are reshaped into patch‑based sequences with positional encodings before attention is applied. The goal of attention is thus to dynamically weight interactions across sequence elements.
>
>
> GMUs, by contrast, are direct analogues of classical computational units such as perceptrons or RBFs. Each GMU has its own weights and biases and produces an activation response, but in the linear case this response is obtained by computing a **$k$‑subspace distance** (Section 3.3) followed by an activation function. This generalizes Euclidean distance, making GMUs a **natural evolution of the RBF** unit. As shown in **Theorem 4, Corollary 4.1, and Proposition 4, $k$‑subspace distance is less prone to the curse of dimensionality**, particularly when data lie on low‑dimensional manifolds; a property common to most natural datasets.
>
>
>
> Therefore, this is not an “either–or” scenario, where it is either GMUs or self-attention, as **they are addressing separate problems**. In fact, there are two separate ways in which GMUs can be quite directly integrated into self-attention based networks.
>
>
> 1. **Replacing the similarity score with $k$-subspace distance.**
> Just as attention computes similarity via query–key dot products: $ \text{score}(q,p) = \frac{q^\top p}{\sqrt{d}},$ recent variants (e.g., **Euclidean/Elliptical attention in Nielsen et al., NeurIPS 2024**) show that the score can be any well-posed comparison metric. A GMU-based score **replaces the dot product with the $k$-subspace distance**:
> $$ score_{k,W}(q,p) = \phi\left( \frac{1}{d} \min_{\theta \in \mathbb{R}^k}
> \left\lVert q-p - W^\top \theta \right\rVert^2 \right),$$
>
> which generalizes Euclidean distance ($k=0$) and is theoretically less prone to the curse of dimensionality on low intrinsic-dimensional manifolds (Proposition 4). Here $W$ is learnable, just like the scaling matrix $M$ in Elliptical attention. This yields an attention mechanism grounded in linear generative subspace projections.
>
> 2. **Substituting MLP layers with GMU layers.**
> Alternatively, GMUs can be integrated into self-attention based networks by just **replacing selected MLP layers** with GMU layers, preserving the attention pipeline while enhancing its feedforward components with subspace-distance based activations. This mirrors the experiments in our work, and in this case can be an effective way to leverage GMUs’ improved out-of-distribution performance without affecting the core attentional aspects.
>
>
> As we have shown in this paper, for linear GMUs the $k$‑subspace distance **admits a closed‑form expression (Eq. (3))** and can be efficiently parallelized using tensorized computation with Einstein summations, enabling **real‑time inference (see Tables 14/15 and https://streamable.com/anox59)**. This makes GMUs practical to deploy in modern architectures. In particular, they can be integrated into attention frameworks either by replacing the similarity score with the $k$-subspace distance, or by substituting selected MLP layers with GMU layers. This is currently an **active direction of future work**.
>
>
>
>
> >  Why is it still necessary to propose and study GMU as universal approximators in this era on top of self-attention?
>
> We believe this point is fully addressed in our previous response, where we contrasted attention and GMUs, explained k‑subspace distance, and outlined two integration pathways (score replacement and MLP substitution).

---

> > ### Author Response · Authors · 2025-12-04
> > **Response-2**
> >
> > > Is it possible to answer the question why foundation models can hallucinate and solve this problem? Can GMU explain and improve the interpretability of foundation models?
> >
> > Given the **better out-of-distribution performance of GMU**, when tested extensively across our 30 synthetic experiment configurations (each of which has an out-of-distribution component) and across 38 total out-of-distribution cases in vision across four datasets (representing various types of corruptions; Tables 4-7), it is very likely that GMU-based foundation models may generalize better and reduce hallucinations. Moreover, we have a specific experiment that precisely tests the **fine-tuning based adaptability of GMUs**, when fine-tuned on the novel out-of-distribution datasets starting from their base data trained model configuration, in Tables 5-7. Called **test-time adaptation (TTA)**, we see that GMU based architectures not only **significantly outperform benchmarks without TTA, but also with TTA**. This shows that the feature representations extracted from GMU architectures are **significantly more conducive for transfer-learning type problems**, and we expect these benefits to extrapolate to the foundation model setting.
> >
> >
> > > Compared with newly proposed computational units such as self-attention, KAN or capsule networks, does GMU demonstrate higher generalizability and robustness theoretically or empirically, especially against OOD scenarios?
> >
> >
> > We have significantly expanded Appendix D to clarify how GMUs differ in theory and motivation from **self‑attention (Appendix D), KANs (Appendix D.1), and capsule networks (Appendix D.2)**. These units are not direct counterparts to GMUs, as their design goals are distinct, though not mutually exclusive. In fact, the core GMU computation, the $k$‑subspace distance, offers a principled alternative to dot‑product operations and could potentially be integrated into self‑attention, KANs, or capsule networks as a future direction.
> >
> > A systematic OOD generalization comparison across all newly proposed computational units would be valuable, but is beyond the scope of this paper. Our focus here is to firmly establish the relevance of GMUs by comparing them with their closest counterparts, which are MLPs and RBFs, through extensive theoretical analysis and rigorous empirical evaluation. This provides a clear foundation for future work exploring how GMUs interact with or extend other emerging architectures.

---

> ### Author Response · Authors · 2025-12-04
> **Response-3**
>
> > For GMU, could the authors give an example of nonlinear GMU? Why in this paper nonlinear GMU is not chosen as a computational unit in deep learning models for study?
>
> Non-linear GMUs can take many forms, but unlike the linear case they lose the closed-form solution (eq. (3)) and **become slower to compute**. Therefore we don’t study them in this work. However, it is quite possible to extend our work to the case where the generative models are non-linear and avoid significant increase in computation time. This is because there exist approaches which can **one-shot approximate solutions to non-linear fitting problems** (Moré 1978, Rahimi, A., & Recht, B. 2007). We thus consider this one of the significant directions of future work with GMUs.
>
> An example of non-linear GMUs would be where the internal generative manifold is a 2nd order polynomial. In this case, the function family $\mathcal{F}$ contains mappings of the form:
> $  f(\theta, W) = W^{(1)} \theta + W^{(2)} (\theta \odot \theta) + b$
> where $\theta \in \mathbb{R}^k$ are the latent generating variables, $W^{(1)} \in \mathbb{R}^{k \times d}$ are the linear weights, $W^{(2)} \in \mathbb{R}^{k \times d}$ are the quadratic weights, and $(\theta \odot \theta)$ denotes the elementwise square of $\theta$. The bias term $b \in \mathbb{R}^{1 \times d}$ shifts the generative manifold.
>
> Thus, the **non-linear second order polynomial GMU** computes:
> $$
> gmu_{2}(X) = φ( (σ^2 d)^{-1} min_{θ} || (X-b)/η - (W^{(1)} θ + W^{(2)} (θ ⊙ θ)) ||^2 )
> $$
>
> Here the generative manifold is quadratic in $\theta$, allowing the GMU to capture richer non-linear structures than the linear case.
>
> Given the above however, a natural question is whether linear GMUs remain useful when the **underlying data manifolds are non-linear**. Our experiments in **Appendix C.3 (sparse neural structure sampling)** and **C.5 (polynomial boundary sampling)** directly address this. In both cases, the data is **generated from explicitly non-linear manifolds, yet linear GMUs consistently outperform baselines**:
>
> - **C.3: Sparse Neural Structure Sampling**
>   Data generated by 2-layer tanh networks yields class-specific non-linear manifolds. Linear GMUs scale better with dataset size and achieve near Pareto-optimal accuracy vs. complexity, outperforming MLP, RBF, and ResNet baselines.
>
> - **C.5: Polynomial Boundary Sampling**
>   Data is separated by polynomial decision boundaries of order 1–3. Linear GMUs (and GMU-MLPs) consistently outperform linear layers and standard MLPs, with gains increasing as polynomial order rises. GMUs generalize better across binary and multi-class settings, often establishing the Pareto front.
>
> **In short, our tests confirm: even when the data manifolds are non-linear, linear GMUs exploit local linearity and yield better fit and generalization while retaining efficiency. This highlights the capability of linear GMUs, and addresses why non-linear GMUs are a natural but not a necessary extension at this point.**
>
>
> #### Moré, J. J. (1978). The Levenberg–Marquardt Algorithm: Implementation and Theory. In Numerical Analysis (Lecture Notes in Mathematics, vol. 630).
>
> #### Rahimi, A., & Recht, B. (2007). Random Features for Large-Scale Kernel Machines. In Advances in Neural Information Processing Systems (NeurIPS).

---

### Official Review · Reviewer_R3fg · 2025-10-31

**Soundness:** 3
**Presentation:** 3
**Contribution:** 3
**Rating:** 8
**Confidence:** 3

**Summary:**

A new computational unit (a kind of layer of a network) is proposed. The idea is to use the distance of a data point from the manifold corresponding to a structural causal model as a feature. The theoretical results presented in the paper show that such a model is reasonable because it can achieve the Bayes optimal classifier when the data come from the structural causal model. The experimental results show the strong generalization capability of the proposed model.

**Strengths:**

The paper is nicely structured. The motivation of the proposed computational unit, theoretical analysis, and the experiments are presented in a well-organized manner.

The idea of the proposed computational unit is reasonable.

The experiments show the strong capability of the proposed model.

**Weaknesses:**

I have no major concerns. Below are minor comments.
- Having a separate section merely for listing the contributions looks strange. It is usually a part of the introduction section. (I even think such a list is of no use though)
- In Eq. (3) (and relevant equations in the appendix), why there is $\cdot^\top$ in the second occasion of $\frac{X-b}\eta$? No transpose seems needed.
- Direct comparison of the numbers of the parameters of the baseline and proposed models would be informative.

**Questions:**

The empirical result in the part of Curse of Dimensionality is not necessarily straightforward to interpret. Especially in Figure 3(a), the CV values of higher dimensionalities (say $d>40$?) seem very close to each other. Does it suggest that such dimensionalities are still hard to deal with? Or does it look so just because of the visualization?

---

> ### Author Response · Authors · 2025-12-04
> **Response**
>
> We thank the reviewer for their comments and suggestions. Below we address their specific questions and concerns.
>
> > Having a separate section merely for listing the contributions looks strange. It is usually a part of the introduction section. (I even think such a list is of no use though)
>
> We have now combined the two sections.
>
>
>
> > In Eq. (3) (and relevant equations in the appendix), why there is $\cdot^\top$ in the second occasion of $\frac{X-b}\eta$? No transpose seems needed.
>
> Yes, the results hold without transpose and we have rectified this. There was some mismatch earlier in describing the $\frac{X-b}\eta$, whether it was to be a column or a row vector. We have now fixed all instances to column vectors for consistency.
>
> > Direct comparison of the numbers of the parameters of the baseline and proposed models would be informative.
>
> We had provided parametric comparisons for all Tabular dataset results (**Table 13**) and most of synthetic experiments where we study performance scalability w.r.t dataset size and model parametric complexity to compare across models (**Figures 4-7**). For completeness, we now also provide the counts for the vision experiments, and compile all parameter counts across all real data experiments in **Appendix F.6**.
>
>
>
>
> > The empirical result in the part of Curse of Dimensionality is not necessarily straightforward to interpret. Especially in Figure 3(a), the CV values of higher dimensionalities (say $d>40$?) seem very close to each other. Does it suggest that such dimensionalities are still hard to deal with? Or does it look so just because of the visualization?
>
>
> In Fig. 3(a), the CV values for higher dimensionalities are close, but for k-subspace distance the **CV is always strictly larger** when zoomed in (also see Corollary 4.1). This does suggest that if the data exists densely within those higher dimensionalities, the CV improvements using $k$-subspace distance are smaller as $d$ increases. However, our follow-up experiment (Fig. 3 (b); Proposition 4 and Remark 5) indicates that when the data in those high ambient dimensionalities **has a lower intrinsic dimension**, $k$-subspace distance can yield very significant gains as $k$ increases, and the **gains persist as data dimensionality increases**. That can be seen in Fig 3(b), when $d=50$, the CV of $k=8$ subspace distance is more than twice the CV of standard Euclidean distance ($k=0$).

---

### Author Response · Authors · 2025-12-04
**General Response**

We have updated the manuscript with the following changes (main):

- Clear reasoning behind the use of GMU in the first layer (raised by reviewers **cDvd** and **zBLM**) has been provided in Section 4 (and the responses to the respective reviewers)
-  The related work section has been significantly expanded to discuss connections and differences with self-attention, Kolmogorov-Arnold Networks and Capsule Networks in much greater detail (**Appendix D, D.1 and D.2**). Theoretical and motivational differences are outlined. This is in response to Reviewer **7BWj**'s main concern.
-  Parameter counts for all real dataset experiments have now been summarized in **Appendix F.6**. This is in response to **R3fg**'s question.

Overall, we found the main concerns were around:

1. **GMUs only used in first layer**: This is standard for RBF-style networks. Please see response to **cDvd**
2. **Comparing and contrasting GMUs with other common computational units (self-attention, KANs, Capsule Networks)**: We hope our extensive discussion in **Appendix D** addresses this point satisfactorily. Mainly, all these units have significantly different motivations and function forms.
3. **Why not try non-linear GMUs**: This is because linear GMUs have one-shot closed-form expressions for the minimization in eq. (1), whereas non-linear GMUs may not. Furthermore, linear GMUs already are a natural point of study and analysis, theory and experiment-wise. Even so, we note that when the underlying data manifolds are non-linear, linear GMUs still show superior fit and generalizability due to its ability to exploit local linearities in the manifolds (**Appendix C.3 and C.5**). More details are in response to **7BWj**

---

### Note · Program_Chairs · 2026-01-17
**Submission Desk Rejected by Program Chairs**

The following references in this submission do not refer to real documents and/or have major errors in bibliographic information:

 Jing Wang, Jiayuan Ding, J Zico Kolter, and Nathan Umbach. Test-time adaptation via self-training. arXiv preprint arXiv:2103.15802, 2021.